# Balancing Utility and Privacy: Dynamically Private SGD with Random Projection

**Zhanhong Jiang**[#]                                                                          *zhjiang@iastate.edu*
**Md Zahid Hasan****                                                                       *zahid@iastate.edu*
**Nastaran Saadati***                                                                     *nsaadati@iastate.edu*
**Aditya Balu**[#]                                                                            *baditya@iastate.edu*
**Chao Liu*****                                                                               *cliu5@tsinghua.edu.cn*
**Soumik Sarkar***[#]                                                                   *soumiks@iastate.edu*

*\*Department of Mechanical Engineering, [#]Translational AI Center, \*\*Department of Electrical and Computer Engineering, Iowa State University*
*\*\*\*Department of Energy and Power Engineering, Tsinghua University*

**Reviewed on OpenReview:** *https://openreview.net/forum?id=u6OSRdkAwl*

## Abstract

Stochastic optimization is a pivotal enabler in modern machine learning, producing effective models for various tasks. However, several existing works have shown that model parameters and gradient information are susceptible to privacy leakage. Although Differentially Private SGD (DPSGD) addresses privacy concerns, its static noise mechanism impacts the error bounds for model performance. Additionally, with the exponential increase in model parameters, efficient learning of these models using stochastic optimizers has become more challenging. To address these concerns, we introduce the Dynamically Differentially Private Projected SGD (D2P2-SGD) optimizer. In D2P2-SGD, we combine two important ideas: (i) dynamic differential privacy (DDP) with automatic gradient clipping and (ii) random projection with SGD, allowing dynamic adjustment of the tradeoff between utility and privacy of the model. It exhibits provably sub-linear convergence rates across different objective functions, matching the best available rate. The theoretical analysis further suggests that DDP leads to better utility at the cost of privacy, while random projection enables more efficient model learning. Extensive experiments across diverse datasets show that D2P2-SGD remarkably enhances accuracy while maintaining privacy. Our code is available here.

## 1 Introduction

Deep learning models Thirunavukarasu et al. (2023); Menghani (2023), enabled by stochastic optimization techniques, have achieved remarkable success in many fundamental machine learning tasks. Though these models empirically show incredibly appealing capabilities, there are critical concerns regarding the privacy of these models. In numerous applications, such as healthcare Chen et al. (2021) and finance Goodell et al. (2021), training datasets often contain highly sensitive information that must remain confidential. However, due to the widespread use of deep learning models, their rich representations can possibly disclose private information under privacy attacks, as demonstrated in the prior works Zhao et al. (2020); Wang et al. (2022a). Additionally, as the number of model parameters increases exponentially, it is unclear to us how privacy and model performance affect each other as the learning becomes more computationally complex.

**Related Works.** To mitigate these privacy concerns, *differential privacy* (DP) Dwork (2006; 2008) was introduced, and it has gained considerable attention Ji et al. (2014); Blanco-Justicia et al. (2022) to provide principled and rigorous privacy guarantees. Intuitively speaking, DP is a mechanism to ensure that all data samples have no significant impact on the ultimate trained model. Differentially private SGD (DPSGD) Abadi et al. (2016); Bassily et al. (2014) is one of the most acknowledged methods to solve the

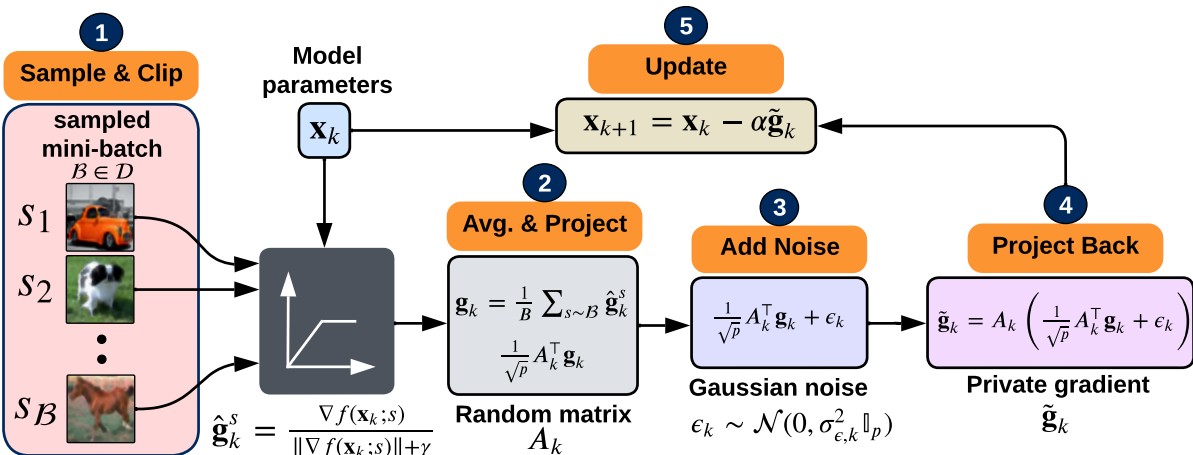

Figure 1: D2P2-SGD method involves five steps. For more technical details, please see Algorithm 1.

private empirical risk minimization (ERM) problems. Specifically, it perturbs each gradient update with a static random noise vector (with the same dimension as that of the gradient) sampled from a distribution. The dimension of the noise vector is typically the same as that of the gradient, which could be extremely large and problematic. More advanced variants on top of DPSGD have also been developed to consider other key issues, such as communication-efficient distributed setting Agarwal et al. (2018); Li & Chi (2025), non-smooth losses Wang et al. (2022b), uniform Lipschitz condition Das et al. (2023), and compilation and vectorization Subramani et al. (2021). Using the perturbed gradient updates, we can compute the tradeoff between the utility and privacy of the model. This tradeoff can be adjusted via a noise mechanism, typically sampled from a *static* distribution with properly chosen yet fixed variance. While this approach is technically simple and provably effective, the noise variance significantly impacts the ultimate error bound. A challenging issue that remains in various private machine learning tasks is how to achieve a desirable tradeoff between privacy and utility, which becomes particularly important in large-scale deep learning models Flemings et al. (2024); Mattern et al. (2022). Recent work Du et al. (2021) proposed a dynamic DP mechanism to adjust the tradeoff on the fly, reducing the model performance loss gap at the cost of increased privacy loss. Further, the dimension of the noise vector is typically the same as that of the gradient, which could be extremely large to cause intensive computational complexity. This issue motivates us to seek out an approach that assists in reducing the complexity while maintaining privacy. Inspired by the work Blocki et al. (2012), we pay attention to model compression techniques as follows. Though numerous prior works have extensively studied the tradeoff, most attempts rarely investigated how to dynamically adjust it along the optimization using a dynamic DP mechanism.

*Model compression* Buciluǎ et al. (2006); Choudhary et al. (2020) has been utilized to reduce the computational complexity, including quantization Chmiel et al. (2020), regularization Moradi et al. (2020); Orvieto et al. (2023), and projection Gu et al. (2023); Tsfadia (2024). Though these methods aim to reduce computational complexity, their technical details can differ significantly depending on the specific focus. For example, in Gu et al. (2023), the authors used projection to identify the dominating gradient subspace, which facilitated improving model accuracy without changing the dimension of model parameters. This is akin to L1-norm regularization Xu et al. (2008), which forces certain model parameters to become exactly zero. Although model compression offers promising performance, it comes at the expense of possible accuracy reduction and sophisticated compression techniques, necessitating an effective optimizer that can balance the dynamic trade-off between privacy and utility. Additionally, the dimension of the noise vector is typically the same as that of the gradient, which could be extremely large and problematic. Consequently, effective approaches should be studied further to look into the tradeoff between privacy and utility while lowering the dimension of the additive noise. Additional related works are provided in Appendix A.1.

In this work, we highlight the need for an effective optimizer to balance among complexity, privacy and utility; answering the critical question:

*Can we design an adaptive differentially private optimizer to allow a small model performance reduction gap and maintain privacy?*

**Contributions.** In this work, we answer the above question affirmatively. Specifically, we propose a novel stochastic optimizer termed Dynamically Differentially Private Projected SGD (D2P2-SGD) (as shown in Figure 1), which, for the first time, integrates the dynamic DP mechanism with automatic gradient clipping and random projection for optimization. The dynamic DP mechanism involves an isotropic Gaussian distribution with a properly chosen *time-varying* variance that decreases along with the iterations, reducing noise effects as privacy loss increases. To further warrant differential privacy and bound the influence of each individual example on the stochastic gradient, we resort to a recently developed automatic gradient clipping mechanism Bu et al. (2024). This is different from the traditional clipping method Abadi et al. (2016), where an upper bound is imposed for gradients. Additionally, the stochastic gradient is projected into a lower-dimensional space, reducing the dimension of additive noise and mitigating the increase in privacy loss. Concretely, the main contributions are as follows:

1. We introduce a novel DP optimizer, D2P2-SGD, that employs a dynamic DP mechanism with a *time-varying* noise variance and random projection. This approach minimizes model performance loss and reduces the dimensionality of noise vectors added to the stochastic gradients. Additionally, the per-sample gradient normalization serves as an automatic gradient clipping mechanism to regulate the influence of individual examples on the stochastic gradient.

2. Theoretically, we prove that D2P2-SGD achieves sub-linear convergence rates for both generally convex and non-convex functions, matching the best available convergence rate of regular SGD. The results for the dynamic DP mechanism can immediately degenerate to those for static scenarios, revealing the consolidation among complexity, utility, and privacy.

3. Extensive evaluations on a wide spectrum of datasets confirm that D2P2-SGD significantly improves model accuracy compared to baseline methods.

While building on prior work Du et al. (2021); Kasiviswanathan (2021), our framework introduces fundamental innovations: (1) The first unified integration of dynamic DP with random projection, where time-varying noise (which is different from the one used in Du et al. (2021)) is explicitly optimized for dimension-reduced gradients, which enables synergistic privacy amplification unaddressed in isolated approaches; (2) Novel theoretical insights into the dimension-privacy-utility trilemma under convex (Theorem 2) and non-convex (Theorem 3) objectives, revealing how projection reshapes dynamic DP trade-offs; (3) A flexible mechanism-agnostic design supporting arbitrary noise/projection variants. Extensive validation confirms these co-adaptations consistently outperform conventional combinations, offering new pathways for efficient privacy-utility balancing. We believe this represents a meaningful conceptual and practical advance in scalable private learning. In this paper, our approach prioritizes theoretical exploration over scalability to larger datasets and models, including transformers and large language models Kasneci et al. (2023). While progress has been made in this domain Yu et al. (2021), designing a differentially private optimizer for such models remains a significant challenge and is deferred to future work.

## 2 Problem Formulation and Preliminaries

Given a private dataset $\mathcal{D} = \{s_1, s_2, ..., s_n\}$ sampled in an i.i.d. manner from a distribution $\mathcal{P}$ such that we want to solve the empirical risk minimization (ERM) problem subject to differential privacy:

$$\min_{\mathbf{x}} f(\mathbf{x}) = \frac{1}{n} \sum_{s \in \mathcal{D}} f(\mathbf{x}, s), \tag{1}$$

where $\mathbf{x} \in \mathbb{R}^d$ and $f(\cdot, \cdot)$ is the loss for a single sample. We aim to optimize Eq. 1 with a gradient-based algorithm in a differentially private manner. We denote by $\mathbf{x}_k$ the model parameters' iterate and $\mathbf{g}_k$ the

Table 1: Comparison among different methods.

| Method | Noise | Compression | Rate |
|---|---|---|---|
| DPSGD[1] | Static | N | $\mathcal{O}(\frac{1}{\sqrt{K}})$ |
| PDP-SGD[2] | Static | Y | $\mathcal{O}(\frac{1}{\sqrt{K}})$ |
| DPKD[3] | Static | N | N/A |
| Anti-PGD[4] | Static | N | $\mathcal{O}(\frac{1}{\sqrt{K}})$ |
| Dynamic DPSGD[5] | Dynamic | N | $\mathcal{O}(\frac{1}{\sqrt{K}})$ |
| PrivSGD[6] | Static | Y | $\mathcal{O}(\frac{1}{\sqrt{K}})$ |
| RQP-SGD[7] | Static | Y | $\mathcal{O}(\frac{1}{\sqrt{K}})$ |
| ADP-SGD[8] | Static | N | $\mathcal{O}(\frac{1}{\sqrt{K}})$ |
| PORTER[9] | Static | Y | $\mathcal{O}(\frac{1}{\sqrt{K}})$ |
| **D2P2-SGD (Convex)** | Dynamic | Y | $\mathcal{O}(\frac{1}{\sqrt{K}} + \frac{\ln K}{K^{1.5}})$ |
| **D2P2-SGD (Non-convex)** | Dynamic | Y | $\mathcal{O}(\frac{1}{\sqrt{K}} + \frac{\ln K}{K^{1.5}})$ |

1: Bassily et al. (2014); 2: Zhou et al. (2020); 3: Mireshghallah et al. (2022); 4: Koloskova et al. (2023b); 5: Du et al. (2021); 6: Kasiviswanathan (2021); 7: Feng & Venkitasubramaniam (2024); 8: Bu et al. (2024); 9: Li & Chi (2025); N: no; Y: yes; $K$: the number of iterations.

mini-batch gradient at each time step $k$. Throughout the analysis, we assume that $\mathbf{g}_k$ is the unbiased estimate of $\nabla f(\mathbf{x}_k)$, i.e., $\nabla f(\mathbf{x}_k) = \mathbb{E}(\mathbf{g}_k)$. In this context, we resort to *gradient clipping mechanism* to constrain the magnitude of the stochastic gradient $\mathbf{g}_k$. To provide the guarantee of differential privacy, it requires bounding the influence of each individual example on $\mathbf{g}_k$. A fairly popular clipping operation Abadi et al. (2016) applied to vector $\mathbf{v} \in \mathbb{R}^d$ is as: $\text{clip}(\mathbf{v}, G) = \min\{1, \frac{G}{\|\mathbf{v}\|}\} \cdot \mathbf{v}$, where $G > 0$, $\|\cdot\|$ is the $l_2$ norm. However, when applying this to the stochastic gradient, such an operation will inevitably result in a "lazy region" issue, particularly if $\|\mathbf{v}\| > G$. This means the parameters will not be updated even if the true gradients are non-zero. Therefore, to mitigate this issue, we leverage a recently developed *per-sample gradient normalization* Bu et al. (2024) as an automatic clipping mechanism described as: $\text{clip}(\mathbf{v}, G, \gamma) = \frac{G}{\|\mathbf{v}\|+\gamma} \cdot \mathbf{v}$, where $\gamma$ is a positive stability constant, which is practically small. Additionally, the authors even showed that any constant choice $G$ is equivalent to choosing $G = 1$. In this work, for the algorithmic framework, we follow their setup to directly set $G = 1$, while still keeping $G$ in the theoretical analysis, particularly for the generally convex objective, which is missing in Bu et al. (2024). Additionally, we will reveal that for any $\gamma > 0$, when the objective is non-convex, the gradient norm will converge to a neighborhood of the optimal solution affected by $\gamma$. To characterize the analysis for the proposed scheme, we introduce the necessary background and preliminary knowledge in the sequel, starting with the standard definition of differential privacy.

**Definition 1.** *(($\varepsilon, \delta$)-differential privacy Dwork (2006)) A randomized algorithm $\mathcal{M}$ is ($\varepsilon, \delta$)-differentially private if for any two neighboring datasets $\mathcal{D}, \mathcal{D}'$ and for all events $\mathcal{Y} \subseteq Range(\mathcal{M})$ in the output range of $\mathcal{M}$, we have $Pr\{\mathcal{M}(\mathcal{D} \in \mathcal{Y})\} \leq \exp(\varepsilon)Pr\{\mathcal{M}(\mathcal{D}' \in \mathcal{Y})\} + \delta$, where the probability is taken over the randomness of $\mathcal{M}$.*

*Range*$(\mathcal{M})$ refers to the set of all possible outcomes of $\mathcal{M}$. Technically speaking, the set $\mathcal{Y}$ in Definition 1 must be measurable. This definition implies that the probability of observing a specific output on any two neighboring datasets can differ by at most a multiplicative factor of $\exp(\varepsilon)$. Intuitively, a sufficiently small $\varepsilon$ value suggests that either including or excluding a single data point from the dataset does not likely affect the output. Hence, an adversary only accessing the output of $\mathcal{M}$ makes it difficult to infer whether any data point is present in the dataset. The parameter $\varepsilon$ is called *privacy budget* and its practical selection varies significantly, depending on different scenarios Ponomareva et al. (2023). Additionally, $\delta$ represents the probability that the privacy guarantee of a differentially private mechanism might be violated and controls the strength of the relaxation, with smaller values leading to stronger privacy guarantees. A generally

recommended $\delta$ value in the literature is to choose $\delta \ll \frac{1}{n}$ Ponomareva et al. (2023). In our analysis, we will establish the privacy guarantee for the proposed algorithm presented in the next section. Before that, we present preliminaries on random projection and formally define the projection matrix. Random projection (RP) Achlioptas (2001) is an effectively fundamental tool that has been used in numerous applications to analyze datasets and then characterize their major features. It projects data points to random directions that are independent of the dataset, which renders simpler and computationally faster trends than classical methods such as singular value decomposition (SVD). RP is based upon the Johnson-Lindenstrauss (JL) lemma Larsen & Nelson (2017) as follows.

**Lemma 1.** *(Johnson-Lindenstrauss Lemma Larsen & Nelson (2017)) For any $0 < \zeta < 1$, a set $\mathcal{S}$ of $m$ points in $\mathbb{R}^d$, and an integer $p > 8(\ln m)/\zeta^2$, there exists a linear map $h : \mathbb{R}^d \to \mathbb{R}^p$, such that $(1-\zeta)\|u-v\|^2 \leq \|h(u)-h(v)\|^2 \leq (1+\zeta)\|u-v\|^2$, for all $u, v \in \mathcal{S}$.*

JL lemma states that a set of points in a high-dimensional space can be projected into a lower-dimensional subspace such that their relative distances are nearly preserved.

While we do not directly apply the above definition in our algorithm design, it motivates and underpins our use of RP for model parameters or gradients Kasiviswanathan (2021). Crucially, RP's foundation in the JL lemma originally conceived for data rests on its guarantee of distance preservation. Without this guarantee, RP would degrade into a lossy and unreliable heuristic. The JL lemma mathematically ensures that geometric fidelity survives dimensionality reduction, enabling scalable and accurate computation. When applying RP to parameters or gradients, our goal remains the same: to find an effective linear map that minimizes information loss. Lemma 1 thus justifies extending RP's application from data to gradients, while highlighting the critical importance of the linear map defined subsequently.

**Lemma 2.** *Let $A$ be a Gaussian random matrix of order $d \times p$, i.e., $A_{ij} \sim \mathcal{N}(0,1)$ and $o$ be any fixed vector in $\mathbb{R}^d$. Define the linear mapping $h(\cdot)$ such that $r = h(o) = \frac{1}{\sqrt{p}}A^\top o$. Thus, $r \in \mathbb{R}^p$ and $r_i = \frac{1}{\sqrt{p}}\sum_j A_{ij}o_j$.*

We notice that each element of $A$ is sampled from the same normal distribution $\mathcal{N}(0,1)$, while we use a slightly different variance, $\sigma_A^2$ instead of 1, in our theoretical analysis for a more generic purpose. Combining Lemma 1 and Lemma 2, with a high probability (at least $1 - \frac{1}{n}$) Johnson et al. (1984), we can maintain the precise model expressivity even if applying RP to project parameters or gradients to a low-dimensional space during updates. This allows us to use RP in our proposed algorithm. We will have a detailed discussion in the following section regarding how RP contributes to our method.

## 3 Algorithm and Main Results

### 3.1 Algorithmic Frameworks

D2P2-SGD is shown in Algorithm 1. Line 4 states the key gradient clipping operation to control the influence of the gradient magnitude. Compared to the clipping mechanism applied in Abadi et al. (2016), we do not need to tune the clipping threshold. In Line 5, a mini-batch stochastic gradient is calculated after the per-sample gradient clipping. In Line 6, the stochastic gradient $\mathbf{g}_k$ is projected to the lower-dimensional space $\mathbb{R}^p$ by using the random projector from Lemma 2 such that we have $\frac{1}{\sqrt{p}}A_k^\top \mathbf{g}_k$, which is followed by adding the noise sampled from a Gaussian distribution with time-varying distribution $\sigma_{\epsilon,k}^2 \mathbb{I}_p$, where $\sigma_{\epsilon,k} = \frac{\sigma_\epsilon}{\sqrt{k}}$. With this, $\epsilon_k$ is *independent* of a high dimension $d$, but dependent on a lower dimension $p \ll d$, which fundamentally reduces the noise. The fact that we resort to the decay of $\frac{1}{\sqrt{k}}$ for the variance is motivated by the same setup as the learning rate in stochastic optimizers Bottou et al. (2018), which manipulates the tradeoff between the convergence speed and optimality. Analogously, $\sigma_{\epsilon,k}^2$ controls the impact of the noise mechanism on the tradeoff between privacy and utility in different phases of the optimization. Since the update for $\mathbf{x}_k$ is operated in the original dimension $\mathbb{R}^d$, we multiply the projected stochastic gradient by $A_k$ to project it back to the original one. It is noted that such an implementation will cause projection errors that impact the error bound (which will be observed in the theoretical analysis). However, similar to Wang et al. (2019b), D2P2-SGD implies more efficient model learning as it has now focused primarily on the subspace in $\mathbb{R}^d$ instead of the whole space. Note that the temporal evolution of $A_k$ is due to its elements being sampled from a constant distribution per iteration. We give a geometric intuition here to facilitate

the understanding of why RP is critical in D2P2-SGD. If we regard gradients as vectors in high-dimensional space, the isotropic noise in $\mathbb{R}^d$ corrupts all directions equally. However, RP projects gradients onto a random low-dimensional subspace where based on Lemma 1, "important" directions of gradients are preserved. This ensures that the noise only corrupts the $p$ compressed dimensions such that the gradient signal still remains strong as the original noise has been defused. To summarize, RP induced by JL Lemma acts as an efficient dimensionality compressor in SGD updates to shrink the injected noise but with gradient preservation.

We claim that D2P2-SGD represents a unified framework over existing methods. When $p = 1$ and $A_1 = A_2 = ... = A_K = I$, D2P2-SGD degenerates to dynamically differentially private SGD (D2P-SGD) Du et al. (2021), though the original approach has another gradient clipping mechanism to prevent dynamic DPSGD from diverging and a different formula for $\sigma_{\epsilon,k}^2$. On top of D2P-SGD, if we set fixed variance for $\epsilon_k$, it becomes DPSGD without any random projection. On the other hand, PrivSGD Kasiviswanathan (2021) can also be obtained if D2P-SGD has a fixed variance with the random projection. However, compared to PrivSGD, which involves an extra optimization to convert from the low-dimensional to high-dimensional spaces, our scheme simply uses $A_k$ to replace the optimization, which significantly attenuates the practical implementation complexity. We also use DP2-SGD (differentially private projected SGD) to represent this case. Please see these two variants in Appendix A.2. We also remark on the additional computational overhead due to RP. $A_k$ is regenerated per iteration by sampling from the same distribution. The total computational overhead incurred is $\mathcal{O}(dp)$. In practice, the implementation is layer-wise so the matrix multiplication is not one-shot with all parameters, mitigating the memory issue.

---

**Algorithm 1** D2P2-SGD

---

1: INITIALIZE: Model parameters $\mathbf{x}_1$, step size $\alpha$, number of epochs $K$, lower dimension $p$, random matrices $A_1, A_2, \ldots, A_K$, mini-batch size $B$, training dataset $\mathcal{D}$, noise sequence $\sigma_{\epsilon,1}^2, \sigma_{\epsilon,2}^2, \ldots, \sigma_{\epsilon,K}^2$, gradient clipping parameter $\gamma$
2: **for** $k = 1, \ldots, K$ **do**
3:     Split the dataset $\mathcal{D}$ into mini-batches of size $B$ and randomly sample one mini-batch $\mathcal{B}$
4:     Compute per-sample clipped gradients: $\hat{\mathbf{g}}_k^s = \frac{\nabla f(\mathbf{x}_k;s)}{\|\nabla f(\mathbf{x}_k;s)\| + \gamma}, s \in \mathcal{B}$
5:     Calculate the mini-batch stochastic gradient: $\mathbf{g}_k = \frac{1}{B} \sum_{s \sim \mathcal{B}} \hat{\mathbf{g}}_k^s$
6:     Project noisy gradient using $A_k$: $\tilde{\mathbf{g}}_k = A_k \left( \frac{1}{\sqrt{p}} A_k^\top \mathbf{g}_k + \epsilon_k \right), \epsilon_k \sim \mathcal{N}(0, \sigma_{\epsilon,k}^2 \mathbb{I}_p)$
7:     Update model parameters: $\mathbf{x}_{k+1} = \mathbf{x}_k - \alpha \tilde{\mathbf{g}}_k$
8: **end for**
9: **return** $\mathbf{x}_K$

---

## 3.2 Main Results

We next show the convergence behavior for our proposed D2P2-SGD, with generally convex and non-convex objective functions, and start with assumptions. All proof is deferred to the appendix.

**Assumption 1.** *(a): $f(\mathbf{x})$ is smooth with modulus $L$ for all $\mathbf{x} \in \mathbb{R}^d$, i.e., for any $\mathbf{x}_1, \mathbf{x}_2 \in \mathbb{R}^d$, we have $\|\nabla f(\mathbf{x}_1) - \nabla f(\mathbf{x}_2)\| \le L\|\mathbf{x}_1 - \mathbf{x}_2\|$; (b) throughout the analysis, the minimum value of the objective $f$ exists and is bounded below, i.e., $f^* := f(\mathbf{x}^*), \mathbf{x}^* = \min_{\mathbf{x} \in \mathbb{R}^d} f(\mathbf{x})$ and $f^* > -\infty$.*

Assumption 1 (a) is generic in many previous works Wang et al. (2019b); Du et al. (2021); Kasiviswanathan (2021) to imply that the variations of gradients along with optimization are bounded above by $L$. Many models, even including deep neural networks, can at least be approximately smooth for the corresponding losses.

**Assumption 2.** *The variance of stochastic gradient $\nabla f(\mathbf{x}, s)$ is bounded above by a constant $\sigma > 0$, i.e., $\mathbb{E}[\|\nabla f(\mathbf{x}, s) - \nabla f(\mathbf{x})\|^2] \le \sigma^2, \forall s \in \mathcal{D}$.*

Assumption 2 is popular when analyzing the convergence behavior of SGD-type algorithms due to the mini-batch sampling (Line 5 in Algorithm 1). In some recent works, the bounded gradient assumption is also leveraged Zhou et al. (2020); Zhang et al. (2023), remaining a strong condition for the analysis. For example, if the objective is in a quadratic form, $f(\mathbf{x}) = \mathbf{x}^\top M \mathbf{x}$, where $M$ is a real symmetric matrix, then $\nabla f(\mathbf{x})$

is in a linear form, which violates such an assumption. The author in Kasiviswanathan (2021) used an extra bounded second moment assumption for gradients besides the bounded variance assumption, though it is weaker than the bounded gradient assumption. We would like to clarify that in D2P2-SGD, the full noise distribution (Line 6 in Algorithm 1) we have added in the update is specifically for differential privacy. Besides, we also quantify the dimension distortion error due to random projection with the variance of the sampling distribution. In the sequel, we start the main result with the privacy guarantee.

**Theorem 1.** *(Privacy) Let Assumption 2 hold. There exist constants $C_1, C_2 > 0$ such that for any $\varepsilon \leq \frac{C_1 B^2 K}{n^2}$, D2P2-SGD is $(\varepsilon, \delta)$-differentially private for any $\delta > 0$, if $\sigma_\epsilon^2 \geq \frac{C_2 B^2 K^2 \ln(1/\delta)}{n^2 \varepsilon^2}$.*

The detailed proof is deferred to Appendix A.3. The core idea of the proof is that at each iteration, Line 6 in Algorithm 1 post-processes the Gaussian noise mechanism that perturbs the stochastic gradient $\mathbf{g}_k$ by adding noise $\epsilon_k$. Subsequently, the sequence of $\{\frac{1}{\sqrt{p}} A_k^\top \mathbf{g}_k + \epsilon_k\}_{k=1}^K$ is released to have a privacy guarantee by following the same privacy proof of Theorem 1 in Abadi et al. (2016). However, the significant difference in our work is that the noise variance is time-varying, i.e., $\sigma_{\epsilon,k}^2$. With the explicit form of noise variance we have defined in this work, i.e., $\sigma_{\epsilon,k}^2 = \frac{\sigma_\epsilon^2}{k}$, it is immediately obtained that $\sigma_{\epsilon,1}^2 > \sigma_{\epsilon,2}^2 > ... > \sigma_{\epsilon,K}^2$. In Wang et al. (2019b) and Abadi et al. (2016), the static variance has the lower bound with respect to some key constants such as $K$ and $G$. Thus, as long as $\sigma_{\epsilon,K}^2 \geq \frac{C_2 K B^2 \ln(1/\delta)}{n^2 \epsilon^2}$, the privacy guarantee is attained. Equivalently, $\sigma_\epsilon^2 \geq \frac{C_2 K^2 B^2 \ln(1/\delta)}{n^2 \varepsilon^2}$ in this context. A constant noise variance throughout optimization risks diminishing the final model's utility. To mitigate this, we employ a decreasing noise variance schedule. This dynamic approach provides essential flexibility, enabling adaptive privacy budgeting that more effectively balances the privacy-utility trade-off compared to static mechanisms. A natural question arises regarding RP's impact on DP guarantees. Critically, the stochasticity inherent in RP does not compromise the DP analysis. This is a direct consequence of DP's fundamental post-processing immunity: any computation applied to the output of a DP mechanism retains the original privacy guarantee. This well-established property that has been validated in prior works Zhou et al. (2020); Kasiviswanathan (2021); Feng & Venkitasubramaniam (2024) ensures our use of RP maintains the rigorous privacy bounds for the core mechanism.

Another observation from Theorem 1 is that the size of mini-batch $B$ has an impact on $\varepsilon$. When $B$ enlarges, $\varepsilon$ has a larger upper bound such that the model performance improves with the cost of privacy, which will be evidently validated in the results section. This also intuitively validates the fact that a larger batch typically improves deep learning model performance. Though the authors in Du et al. (2021) for the first time proposed to leverage dynamic DP mechanism to reduce the model performance loss gap, privacy guarantee has been ensured by the dynamic power following an exponential mechanism $\sigma_{\epsilon,k} \propto \mathcal{O}(\rho^{-\frac{k}{K}})$, where $\rho$ is a positive constant. As they still utilized the clipping mechanism from Abadi et al. (2016), they also had to establish a similar exponential mechanism for the clipping threshold, which makes their algorithm framework more complex. While in our work, thanks to the automatic clipping mechanism, there is no such requirement. We are now ready to state the results for the utility with different functions.

**Theorem 2.** *(Utility for convex functions) Let Assumptions 1 and 2 hold. Suppose that $f$ is a convex function and that $A$ is a random matrix with each element being sampled from a normal distribution $\mathcal{N}(0, \sigma_A^2)$. Also, let the additive noise of DP mechanism have the variance $\sigma_{\epsilon,k}^2$. If the step size $\alpha \leq \frac{1}{2L}$, then for the iterates $\{\mathbf{x}_k\}_{k=1}^K, K \geq 1$ generated by D2P2-SGD, the following relationship holds true*

$$\mathbb{E}[f(\bar{\mathbf{x}}_K) - f^*] \leq \frac{\|\mathbf{x}_1 - \mathbf{x}^*\|^2 (1+\gamma)}{2\alpha K \sigma_A^2 \sqrt{p}} + \frac{\alpha p^{1.5}(1+\gamma)(\ln K + 1)\sigma_\epsilon^2}{2K} + \frac{\alpha p d^2 \sigma_A^2 (1+\gamma)}{2}, \quad (2)$$

*where $\bar{\mathbf{x}}_K = \frac{1}{K}\sum_{k=1}^K \mathbf{x}_k$.*

Theorem 2 suggests that the error bound involves three terms: the initialization error, the error of additive noise due to the DP mechanism, and the random projection approximation error. We first turn to the second term and see how it impacts the error bound. If $\sigma_{\epsilon,k}^2 = \sigma_\epsilon^2$, this term becomes $(\alpha p^{1.5}(1+\gamma)\sigma_\epsilon^2)/2$, which is a constant. Instead, if $\sigma_{\epsilon,k}^2 = \frac{\sigma_\epsilon^2}{k}$, it can be bounded by $\mathcal{O}(\frac{\ln K}{K})$, which also relaxes the dependence on $\alpha$ to decay the magnitude. Though the model performance loss gap is reduced, dynamic variance can breach privacy. To maintain the $(\varepsilon, \delta)$-differential privacy for D2P2-SGD, as implied in Theorem 1, the additive

noise should be sampled with a larger $\sigma_\epsilon^2$ to offset the privacy loss, particularly in the early phase during the optimization, compared to DPSGD. When $\sigma_\epsilon \propto (\frac{1}{\varepsilon})$, we can immediately know that with a higher privacy (which corresponds to a smaller $\varepsilon$), the utility decreases. In this scenario, RP becomes more advantageous because noise reduction with a small $p$ counters strict privacy. Conversely, with a lower privacy (which corresponds to a larger $\varepsilon$), the utility increases. However, RP may hurt utility as the dimension distortion error dominates if the noise is already small. Thus, prioritizing one forces tradeoffs in the others, making dimension $p$ the pivotal lever in this trilemma. We include the details of how privacy budget $\varepsilon$ and dimensions $(d, p)$ impact the error bound and the comparison to DPSGD in Appendix A.5. The last term is associated with RP approximation error, while the term $\sigma_A^2 d^2$ due to model compression can cause significant error. One empirical remedy is to leverage a small $\alpha$, but this leads to slow convergence. Eq. 2 implies that when $K \to \infty$, D2P2-SGD converges to the neighborhood of $\mathbf{x}^*$ asymptotically with the rate of $\mathcal{O}(1/K + \ln K/K)$, up to a constant $\alpha p d^2 \sigma_A^2 (1 + \gamma)/2$. Additionally, a small initialization error $\|\mathbf{x}_1 - \mathbf{x}^*\|$ and a larger batch size lead to better performance, which stresses the implication from Theorem 1. Overall, the consolidation among complexity, utility, and privacy in D2P2-SGD is reflected explicitly in Theorem 2. The following corollary summarizes the explicit convergence rate when $\alpha = \mathcal{O}(\frac{1}{\sqrt{K}})$.

**Corollary 1.** *With conditions defined in Theorem 2, when $\alpha = \mathcal{O}(\frac{1}{\sqrt{K}})$, the following relationship holds true, i.e., $\mathbb{E}[f(\bar{\mathbf{x}}_K) - f^*] \leq \mathcal{O}(\frac{1}{\sqrt{K}} + \frac{\ln K}{K^{1.5}})$.*

The conclusion in Corollary 1 requires the constant learning rate $\alpha$ to have a format $\alpha \propto \mathcal{O}(\frac{1}{\sqrt{K}})$, which is a quite popular choice in stochastic optimization Garrigos & Gower (2023). After carefully reviewing some recent works Koloskova et al. (2023a); Bu et al. (2024); Xiao et al. (2023); Chen et al. (2020), though they have also focused on the investigation of convergence guarantee for differentially private SGD with gradient clipping, all of them paid only attention to nonconvex cases. Our result explicitly and rigorously establishes the convergence rate of D2P2-SGD for convex functions. Without dynamic differential privacy and model compression, our result resembles exactly the same convergence rate of regular SGD in $\mathcal{O}(\frac{1}{\sqrt{K}})$. Particularly, if a desired accuracy $\zeta > 0$ is defined specifically for $\mathbb{E}[f(\bar{\mathbf{x}}_K) - f^*] \leq \zeta$, the complexity bound for $K$ is $\mathcal{O}(\frac{1}{\zeta^2} + \frac{(\ln(\frac{1}{\zeta}))^{2/3}}{\zeta^{2/3}})$. In comparison, our rate improves the one reported in Kasiviswanathan (2021), which is $\mathcal{O}(\frac{\ln K}{\sqrt{K}})$. Note that in their case, they use a decaying step size $\alpha_k \propto \frac{1}{\sqrt{k}}$ and there is no smoothness assumption. We will now present the main result for non-convex functions.

**Theorem 3.** *(Utility for non-convex functions) Let Assumptions 1 and 2 hold. Suppose that $A$ is a random matrix with each element being sampled from a normal distribution $\mathcal{N}(0, \sigma_A^2)$. Also, let the additive noise of the DP mechanism have the variance $\sigma_{\epsilon,k}^2$. If the step size $\alpha \leq \frac{1}{2L}$, then for the iterates $\{\mathbf{x}_k\}_{k=1}^K, K \geq 1$ generated by D2P2-SGD, the following relationship holds true*

$$\min_{k \in [1,K]} \mathbb{E}[\|\nabla f(\mathbf{x}_k)\|] \leq \frac{f(\mathbf{x}_1) - f^*}{K \sigma_A^2 \sqrt{p}\alpha} + \frac{\alpha L p^{1.5} \sigma_\epsilon^2 (\ln K + 1)}{K} + L\alpha\sqrt{p}d^2\sigma_A^2 + \frac{2\sigma}{\sqrt{B}} + \gamma. \tag{3}$$

We immediately have the following result when the step size satisfies a certain condition.

**Corollary 2.** *With conditions defined in Theorem 3, when $\alpha = \mathcal{O}(\frac{1}{\sqrt{K}})$, the following relationship hold true, $\min_{k \in [1,K]} \mathbb{E}[\|\nabla f(\mathbf{x}_k)\|] \leq \mathcal{O}(\frac{1}{\sqrt{K}} + \frac{\ln K}{K^{1.5}} + \sigma + \gamma)$.*

Similarly, the error bound from Theorem 3 is dictated by the initialization error, the error of addictive noise, the random projection approximation error and the clipping bias. Especially, the clipping bias $\frac{2\sigma}{\sqrt{B}} + \gamma$ is determined by the variance, the batch size and the stability constant. To mitigate the clipping bias, one immediate solution is to properly increase the batch size. Due to $\gamma$, in practical implementation, one can choose a small value for $\gamma$. When $\gamma$ is $\mathcal{O}(\frac{1}{\sqrt{K}})$ in Corollary 2, the clipping bias resembles the result in Koloskova et al. (2023a), up to some absolute constants. Also, the convergence rate of non-convex functions remains similar to that of generally convex functions. However, comparing results from Corollary 1 and Corollary 2, the error of non-convex functions is larger due to the clipping bias, which illustrates that the convergence for non-convex functions is more challenging. We summarize the clipping bias for different methods in Table 4 in Appendix A.7 for comparison (we only compare for non-convex functions as most

of the existing methods only discussed non-convex objectives). Practically, non-asymptotic convergence is preferred such that we can resort to a pre-defined small constant $\xi > 0$ to define metric, i.e., the norm of gradient $\|\nabla f(\mathbf{x})\| \leq \xi$ for non-convex functions. Suppose that $B = \sigma^2/\xi^2$ and $\gamma = \mathcal{O}(\frac{1}{\sqrt{K}})$. If the computational complexity is defined as the total number of gradient computations, it can be observed that the computational complexity is $KB = \mathcal{O}(1/\xi^4)$, which retains the same complexity as in Ghadimi & Lan (2013). The tradeoff between privacy and utility is analogous to that for convex functions, only differing in some absolute constants. Thus, we do not repeat the same remarks in this context, but provide the impact of privacy budget $\varepsilon$ and dimensions $(d, p)$ on the error bound in Appendix A.8.

## 4 Numerical Experiments

We present extensive empirical results to thoroughly validate our proposed approaches with a comparison to baselines. The baselines we use in this study consist of SGD, vanilla DPSGD, D2P-SGD, and DP2-SGD. D2P-SGD is an equivalent alternative to Dynamic DPSGD in Du et al. (2021). DP2-SGD can also be regarded as an equivalence of PrivSGD Kasiviswanathan (2021) since the compression technique they adopted is also random projection, with a static DP. We leverage the Opacus library Yousefpour et al. (2021) and build the framework on top of it. We use a 4-layer Convolutional Neural Network (CNN) as the model, which has been widely used in testing optimizers. A more detailed explanation of the architecture is provided in Appendix A.9. Additionally, the datasets for testing our algorithms include FashionMNIST and SVHN Figueroa (2019). As we have particularly identified the critical relationship between the privacy loss $\varepsilon$ and other parameters, an ablation study on this is shown to reveal their impacts on the performance. Hyperparameters have a significant impact on deep learning models' performance. In the interest of brevity, we have chosen most hyperparameters used in the model via manual tuning, although we note that an automated hyperparameter-tuning framework would likely be beneficial for these models. Additional information, including network architecture and specific hyperparameter setup, and results on other *larger* models and datasets are in Appendix A.9.

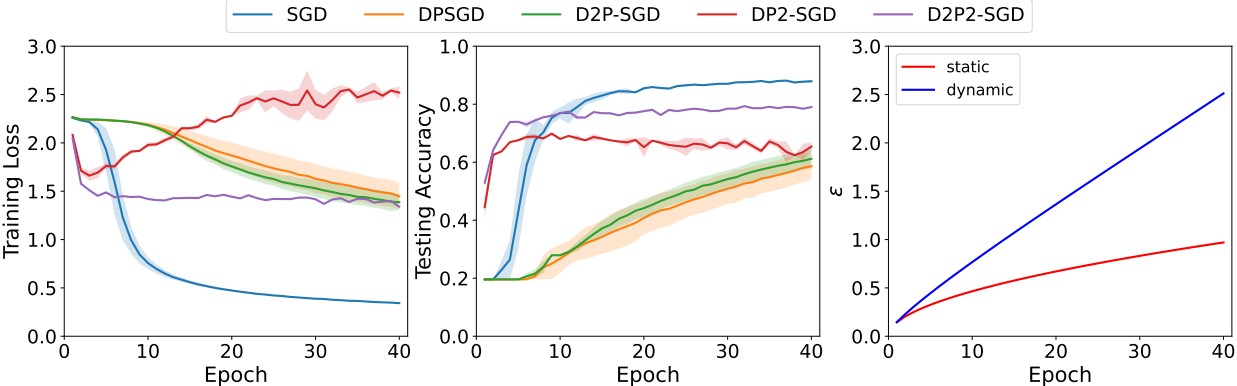

Figure 2: Comparison among methods for SVHN data: on the right side, the privacy loss is shown for static and dynamic scenarios.

**Comparative Evaluation.** Figure 2 shows the model performance and privacy loss for different methods. We train five different instances of each algorithm with different random seeds. The solid curves correspond to the mean and the shaded region to the minimum and maximum values over the five runs. For the privacy loss, the standard deviation is fairly small. Also, the dynamic mechanism for D2P-SGD and D2P2-SGD is the same. Similarly, DPSGD and DP2-SGD have the same static mechanism. According to Figure 2, D2P2-SGD significantly improves the model accuracy compared to DPSGD, D2P-SGD, and DP2-SGD. While this comes at the expense of a larger privacy loss, which is expected. This is attributed to the decreasing variance $\sigma^2_{\epsilon,k}/k$ along with iterations. However, the testing accuracy of D2P2-SGD is much closer to SGD, while having a gap due to projection error and clipping bias. Notably, D2P2-SGD spends a lesser number of epochs, reaching a higher accuracy at the early phase, even earlier than SGD, which implies that

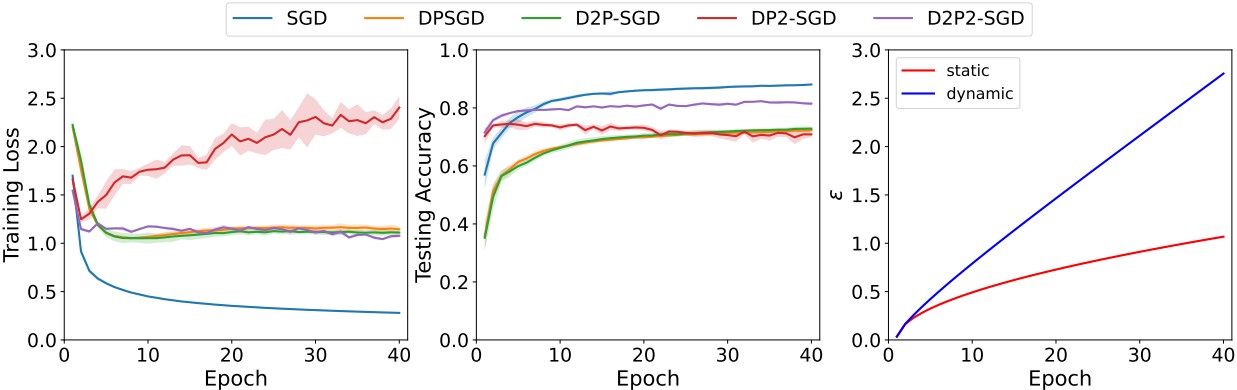

Figure 3: Comparison among methods for FashionMNIST data: on the right side, the privacy loss is shown for static and dynamic scenarios.

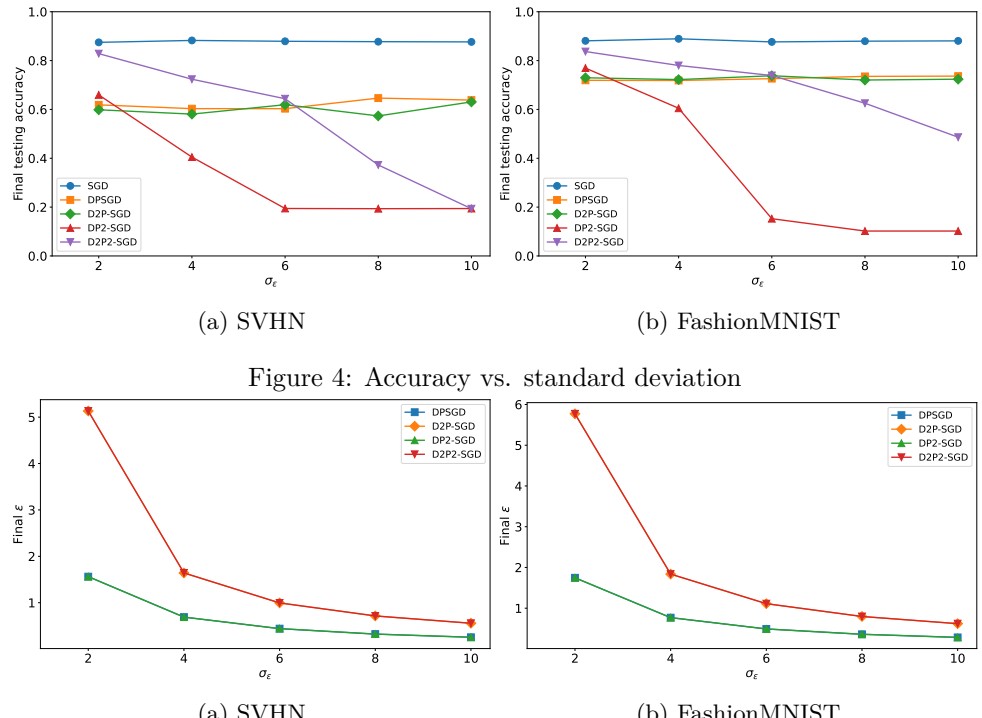

(a) SVHN

(b) FashionMNIST

Figure 4: Accuracy vs. standard deviation

(a) SVHN

(b) FashionMNIST

Figure 5: Privacy loss vs. standard deviation

random projection enables more efficient model learning. Similar conclusions can be made from Figure 3. Comparing DP2-SGD and D2P2-SGD, we see that with random projection, if the noise variance from the DP mechanism is static, the performance deteriorates accordingly. However, with a dynamic mechanism, it performs robustly throughout training. This validates the conclusion from Theorem 3, where the second term decays faster when the number of iterations increases. However, DP2-SGD remains with the rate of $\mathcal{O}(1/\sqrt{K})$ (If $\alpha$ is not in $\mathcal{O}(1/\sqrt{K})$, this term in D2P2-SGD is in $\mathcal{O}(\ln(K/K))$, while DP2-SGD $\mathcal{O}(1)$). Figures 2 and 3 reveal comparable performance between DPSGD and D2P-SGD. This alignment stems from our privacy configuration. In DPSGD, the fixed noise variance $\sigma_\epsilon^2 = \frac{C_2 K \ln(1/\delta) B^2}{n^2 \varepsilon^2}$ yields a static privacy-utility tradeoff. Conversely, D2P-SGD employs dynamic noise variance $\sigma_{\epsilon,k}^2 = \frac{C_2 K^2 \ln(1/\delta) B^2}{n^2 \varepsilon^2 k}$. During early optimization ($k \ll K$), initialization error dominates, masking performance differences despite divergent noise mechanisms. As $k \to K$, privacy error dominates, but the noise variances converge, explaining the sustained similarity in performance. Turning to the privacy loss ($\varepsilon$) in both figures, we can observe that the

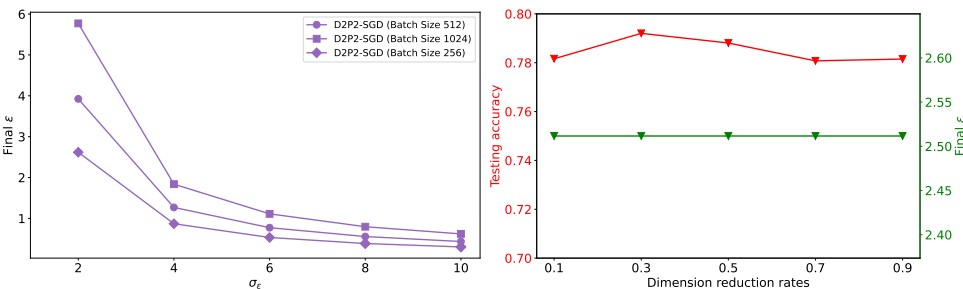

(a) Batch size vs. privacy loss for FashionM-NIST

(b) Dimension vs. accuracy/privacy for SVHN

Figure 6: Impacts of parameters in D2P2-SGD

maximum privacy losses of the dynamic mechanism end up with respectively 2.45 (for SVHN) and 2.75 (for FashionMNIST). Compared to values with the static DP mechanism (1.06 and 0.95, respectively), the privacy loss of D2P2-SGD grows sharply. However, given the bound for $\varepsilon$ in Theorem 1, as long as the constant $C_1$ ($\geq 314$, see Appendix A.9) is selected properly, D2P2-SGD still remains $(\varepsilon, \delta)$-differentially private. While the value of 2.45 or 2.75 offers a reasonable, rather than exceptionally strong privacy guarantee, it reflects our focus on balancing privacy and utility. However, in highly private domains such as medical applications, D2P2-SGD can still be adopted to bias towards privacy by setting large dynamic noise. These two values are the only resulting values based on the data we have applied in this work. Thus, our proposed scheme improves the accuracy over baselines while successfully maintaining differential privacy. Table 2 summarizes the performance for different optimizers at the same privacy loss $\varepsilon$ values ($\sigma_\epsilon = 3.0$), showing that D2P2-SGD outperforms other differentially private optimizers and validates our theoretical finding.

Table 2: Testing accuracy (averaged through 4 random seeds) at different $\varepsilon$ values (FashionMNIST dataset)

| Optimizers | Test acc. at $\varepsilon = 1.0$ | Test acc. at $\varepsilon = 2.0$ | Test acc. at $\varepsilon = 4.0$ | Test acc. at $\varepsilon = 6.0$ | Test acc. at $\varepsilon = 8.0$ |
|---|---|---|---|---|---|
| DPSGD | 0.6910 | 0.5291 | 0.5331 | 0.5704 | 0.5849 |
| D2P-SGD | 0.6549 | 0.7063 | 0.5242 | 0.5656 | 0.5573 |
| DP2-SGD | 0.6042 | 0.6736 | 0.5884 | 0.5461 | 0.6297 |
| D2P2-SGD | **0.7132** | **0.7322** | **0.7417** | **0.7851** | **0.7566** |

**Impact of $\sigma_\epsilon$.** From Figure 4, it shows that as $\sigma_\epsilon$ value increases, the final model accuracy drops for DP2-SGD and D2P2-SGD. This first validates the coupling between projection error $\sigma_A^2$ and the noise variance $\sigma_{\epsilon,k}^2$ in Theorem 3 and shows the trade-off between the utility and privacy. When $\sigma_\epsilon < 6$, which can be treated as a low privacy regime, D2P2-SGD maintains better performance than DPSGD and D2P-SGD, while underperforming in the high privacy regime after $\sigma_\epsilon > 6$. This intuitively shows us a careful selection of $\sigma_\epsilon$ is required to balance the trade-off. Figure 5 delivers a similar conclusion in terms of privacy loss.

**Impact of $B$.** As suggested by Theorem 1, we can adjust the privacy loss by setting the batch size $B$. In Figure 6a, we can observe that when batch size increases from 256 to 1024, it increases the upper bound for $\varepsilon \leq \frac{C_1 B^2 K}{n^2}$ such that the privacy loss is relatively higher through all $\sigma_\epsilon$ for D2P2-SGD, leading to the better performance. This essentially validates the condition $B = \sigma^2/\xi^2$, where $\sigma^2$ decreases in the error bounds, leading to better convergence due to the smaller $\xi$.

**Impact of $p$.** Figure 6b shows the impact of different lower dimensions on the testing accuracy and privacy loss. It immediately suggests that the performance of random projection varies with different $p$ values. The optimal one is a 30% reduction rate, which implies that random projection can assist in model learning efficiency if the $p$ value is chosen properly. Instead, the privacy loss is independent of the dimension change based on Figure 6b. Thus, D2P2-SGD allows us to reduce the computational complexity by random projection while maintaining privacy. The dimension can be chosen in a wide range and will not affect model performance significantly, including accuracy and privacy.

**Limitation.** Though D2P2-SGD has shown good performance compared to the existing baselines, some potential limitations exist, which can also help us close such gaps in future work. First, D2P2-SGD may not work well in scenarios with *high privacy restrictions*. As we have the decaying noise variance that ensures decent model performance, privacy loss will inevitably be the resulting outcome. One can carefully tune these parameters to obtain acceptable values, but it is still fairly challenging to scale. One way to get rid of this is to develop more effective dynamic DP mechanisms such that the tradeoff between utility and privacy can be handled better. Second, getting an optimal $p$ value for random projection may be difficult as well. Though based on Johnson-Lindenstrauss Lemma, $p$ can be analytically obtained, its practical values for different scenarios have not yet been accessible in a principled manner.

## 5 Conclusions and Broader Impacts

This work presents a novel differentially private optimizer termed D2P2-SGD with a dynamic DP mechanism, automatic gradient clipping, and model compression, which reveals the synthesis among privacy, utility, and complexity. Specifically, we establish the dynamic privacy guarantee such that a relatively larger variance is required in the early phase of optimization to compensate for the privacy loss in the latter phase. Given a pre-defined dynamic variance, D2P2-SGD enables a slightly tighter error bound compared to vanilla DPSGD with a static DP mechanism. Empirical results are shown to validate the theory first and then compared with baselines using popular models and benchmark datasets. Compared to vanilla DPSGD, our D2P2-SGD is more robust against larger noise variance but with a slightly larger privacy loss. However, the accuracy is significantly improved without dependence on the large dimension. The broader vision of this work is to advance the field of differentially private machine learning with the potential impact of building privacy-aware deep learning models with highly sensitive information for critical sectors such as healthcare and national security.

## Acknowledgments

We sincerely thank the anonymous reviewers for their thorough evaluation and insightful recommendations, which have helped improve the presentation and clarity of this work. This work was supported by the CO-ALESCE: COntext Aware LEarning for Sustainable CybEr-Agricultural Systems (CPS Frontier #1954556).

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

# A   Appendix

## A.1   Additional Related Works

Beyond the works presented in the main content, differential privacy with dimension reduction has received considerable attention recently due to the emerging data-centric deep learning techniques, which require privacy-preserving yet efficient learning. Inspired by DPSGD Bassily et al. (2014), Zhou et al. Zhou et al. (2020) proposed projected DPSGD performing noise reduction by projecting the noisy gradients to a low-dimensional subspace, which is induced by the top gradient eigenspace on a small public dataset. Though PDP-SGD is theoretically and empirically validated well, the adoption of a public dataset may not be feasible in some scenarios, particularly those being data privacy-critical. Another work Mireshghallah et al. (2022) developed a technique called differentially private model compression specifically for pre-trained large language models such as GPT-2, leveraging a knowledge distillation algorithm. However, the lightweight model comes at the cost of accuracy loss. To mitigate this issue, the authors developed the differentially private iterative magnitude pruning, which produces compressed models whose performance is comparable to the original models. Similarly, the author from Kasiviswanathan (2021) found that with a constraint set, SGD can be operated with the lower-dimensional (compressed) stochastic gradients, and applied it to differentially private learning with nonconvex functions to improve error bounds. However, the additive noise mechanism is still static such that the model performance gap cannot be shrunk along with iterations. In terms of

differential privacy, the authors in Koloskova et al. (2023b) proposed a new stochastic optimization method that is coupled with linearly correlated noise and showed explicit convergence rates in both convex and non-convex objective functions. They also devised a new objective for the offline matrix factorization, improving the convergence property. Du et al. Du et al. (2021) presented for the first time a dynamic differentially private mechanism for SGD that can adaptively trade off between the utility and the privacy. However, their additive noise mechanism involving an exponential function could be sophisticated to implement. A more recent work Feng & Venkitasubramaniam (2024) proposed a novel differentially private optimizer by combining noisy SGD and randomized quantization, which is close to our work. They theoretically analyzed the utility-privacy tradeoff of the proposed scheme on specifically convex losses and empirically showed results regarding the impact of different randomized quantization parameters. Unfortunately, there is no reported result on the nonconvex objectives using the developed method.

## A.2 Additional Algorithms

In this section, we present additional algorithm frameworks degenerated from D2P2-SGD, including D2P-SGD (in Algorithm 2) and DP2-SGD (in Algorithm 3). We also list out the differences among these three algorithms in Table 3.

---

**Algorithm 2** D2P-SGD

---

1: **Input:** Model initialization $\mathbf{x}_1$, step size $\alpha$, the number of epochs $K$, size of mini-batch $B$, training set $\mathcal{D}$, noise sequence $\sigma_{\epsilon,1}^2, \sigma_{\epsilon,2}^2, ..., \sigma_{\epsilon,K}^2, \gamma$
2: **for** each $k$ in 1 to $K$ **do**
3:     Split the dataset $\mathcal{D}$ into multiple mini-batches with size $B$ and randomly sample one $\mathcal{B}$
4:     Clip the per-sample gradient $\hat{\mathbf{g}}_k^s = \nabla f(\mathbf{x}_k; s)/(\|\nabla f(\mathbf{x}_k; s)\| + \gamma), \ s \in \mathcal{B}$
5:     Calculate the mini-batch stochastic gradient $\mathbf{g}_k = \frac{1}{B} \sum_{s \sim \mathcal{B}} \hat{\mathbf{g}}_k^s$
6:     Perturb the gradient $\mathbf{g}_k$ using dynamic noise: $\tilde{\mathbf{g}}_k = \mathbf{g}_k + \epsilon_k$, where $\epsilon_k \sim \mathcal{N}(0, \sigma_{\epsilon,k}^2 \mathbb{I}_p)$
7:     Update parameter using projected noisy gradient: $\mathbf{x}_{k+1} = \mathbf{x}_k - \alpha \tilde{\mathbf{g}}_k$
8: **end for**
9: **Output:** $\mathbf{x}_K$

---

**Algorithm 3** DP2-SGD

---

1: **Input:** Model initialization $\mathbf{x}_1$, step size $\alpha$, the number of epochs $K$, lower dimension $p$, random matrices $A_1, A_2, ..., A_K$, size of mini-batch $B$, training set $\mathcal{D}$, noise variance $\sigma_{\epsilon}^2, \gamma$
2: **for** each $k$ in 1 to $K$ **do**
3:     Split the dataset $\mathcal{D}$ into multiple mini-batches with size $B$ and randomly sample one $\mathcal{B}$
4:     Clip the per-sample gradient $\hat{\mathbf{g}}_k^s = \nabla f(\mathbf{x}_k; s)/(\|\nabla f(\mathbf{x}_k; s)\| + \gamma), \ s \in \mathcal{B}$
5:     Calculate the mini-batch stochastic gradient $\mathbf{g}_k = \frac{1}{B} \sum_{s \sim \mathcal{B}} \hat{\mathbf{g}}_k^s$
6:     Project noisy gradient using $A_k^\top$: $\tilde{\mathbf{g}}_k = A_k(\frac{1}{\sqrt{p}} A_k^\top \mathbf{g}_k + \epsilon_k)$, where $\epsilon_k \sim \mathcal{N}(0, \sigma_{\epsilon}^2 \mathbb{I}_p)$
7:     Update parameter using projected noisy gradient: $\mathbf{x}_{k+1} = \mathbf{x}_k - \alpha \tilde{\mathbf{g}}_k$
8: **end for**
9: **Output:** $\mathbf{x}_K$

---

Table 3: D2P2-SGD and its different variants.

| Method | $p$ | $\sigma_{\epsilon,k}^2$ | $A_k$ |
|---|---|---|---|
| D2P2-SGD | $\geq 1$ | $\frac{\sigma_\epsilon^2}{\sqrt{k}}$ | $\mathbb{R}^{d \times p}$ |
| D2P-SGD | N/A | $\frac{\sigma_\epsilon^2}{\sqrt{k}}$ | $I \in \mathbb{R}^{d \times d}$ |
| DP2-SGD | $\geq 1$ | $\sigma_\epsilon^2$ | $\mathbb{R}^{d \times p}$ |

### A.3 Proof for privacy guarantee

In this subsection, we provide proof of the privacy guarantee. Before that, we present some existing results to characterize the proof. We begin with a function defined in the following.

**Definition 2.** *Denote by $\mathcal{M}$ a randomized mechanism and by $\mathcal{D}$ and $\mathcal{D}'$ two adjacent inputs. The parameterized Rényi divergence between two distributions is defined as:*

$$
\begin{aligned}
e_{\mathcal{M}}(\eta) &= \sup_{\mathcal{D},\mathcal{D}'} D_\eta(\mathcal{M}(\mathcal{D})||\mathcal{M}(\mathcal{D}')) \\
&= \sup_{\mathcal{D},\mathcal{D}'} \frac{1}{\eta-1} \log \mathbb{E}_{\theta \sim \mathcal{M}(\mathcal{D}')}\left[\left(\frac{\mathcal{M}(\mathcal{D})(\theta)}{\mathcal{M}(\mathcal{D}')(\theta)}\right)^\eta\right],
\end{aligned}
\tag{4}
$$

*where $\mathcal{M}(\mathcal{D})(\theta)$ refers to the density at $\theta$ of this distribution.*

With the above definition, the first result as follows is from Theorem 2.1 in Abadi et al. (2016).

**Lemma 3.** *Let $\mathcal{M} = \mathcal{M}_K \circ \mathcal{M}_{K-1} \circ \cdots \circ \mathcal{M}_1$ be defined in an interactively compositional way, then for any fixed $\eta \geq 1$,*

$$
e_{\mathcal{M}}(\eta) \leq \sum_{k=1}^K e_{\mathcal{M}_i}(\eta).
\tag{5}
$$

*$e_{\mathcal{M}}(\eta)$ is defined as in Definition 2.*

*Proof.* The proof directly follows from the proof of Theorem 2.1 (Composability) in Abadi et al. (2016). □

The $K$ iterations of D2P2-SGD can be decomposed into $K$ compositions of sub-sampled Gaussian mechanisms with uniform sampling without replacement. We denote by $\mathcal{G}(\sigma_\epsilon) \circ \mathcal{S}(n, B)$. With the abuse of notations, we denote by $\tilde{e}(\eta)$ the privacy-accountant functional of Eq. 4 in Definition 2. Another lemma in the following introduces a sufficient and necessary condition for a mechanism to be $(\varepsilon, \delta)-$ differentially private.

**Lemma 4.** *Let $\mathcal{M}$ be a randomized mechanism. $\mathcal{M}$ is $(\varepsilon, \delta)-DP$ if and only if $\delta \geq \exp[(\eta-1)(e_{\mathcal{M}}(\eta)-\varepsilon)]$, for some $\eta > 1$.*

*Proof.* Let $P = \mathcal{M}(x)$ and $Q = \mathcal{M}(x')$ for adjacent datasets $x, x'$ and define the privacy loss random variable $L(\zeta) = \log \frac{Pr[P=\zeta]}{Pr[Q=\zeta]}$. We first show the sufficiency, i.e., if the condition holds, then $\mathcal{M}$ is $(\varepsilon, \delta)$-DP. Based on Definition 2, we know that

$$
\exp[(\eta-1)e_{\mathcal{M}}(\eta)] = \mathbb{E}_{\zeta \sim Q}\left[e^{(\eta-1)L(\zeta)}\right].
\tag{6}
$$

We then apply Markov's inequality to the privacy loss under $P$ such that

$$
Pr_{\zeta \sim P}\left[L(\zeta) > \varepsilon\right] = Pr_{\zeta \sim P}\left[e^{(\eta-1)L(\zeta)} > e^{(\eta-1)\varepsilon}\right] \leq \frac{\mathbb{E}_{\zeta \sim P}\left[e^{(\eta-1)L(\zeta)}\right]}{e^{(\eta-1)\varepsilon}}.
\tag{7}
$$

Transforming the expectation under $P$ to one under $Q$ yields

$$
\mathbb{E}_{\zeta \sim P}\left[e^{(\eta-1)L(\zeta)}\right] = \mathbb{E}_{\zeta \sim Q}\left[e^{\eta L(\zeta)}\right] = \exp[(\eta-1)e_{\mathcal{M}}(\eta)].
\tag{8}
$$

Substituting Eq. 8 to Eq. 7 and simplifying it grants us the following

$$
Pr_{\zeta \sim P}\left[L(\zeta) > \varepsilon\right] \leq \exp[(\eta-1)e_{\mathcal{M}}(\eta)] \leq \delta.
\tag{9}
$$

The second inequality is based on the condition $\delta \geq \exp[(\eta-1)(e_{\mathcal{M}}(\eta)-\varepsilon)]$. Thus, this implies $Pr[L > \varepsilon] \leq \delta$, which is equivalent to $(\varepsilon, \delta)$-DP. We now show the necessity, i.e., If $\mathcal{M}$ is $(\varepsilon, \delta)$-DP, then the condition holds

for some $\eta > 1$. By the definition of $(\varepsilon, \delta)$-DP, the privacy loss $L$ satisfies $Pr_{\zeta \sim P}[L(\zeta) > \varepsilon] \leq \delta$. Then, we have for any $c > 1$

$$\exp[(c-1)e_{\mathcal{M}}(c)] = \mathbb{E}_{\zeta \sim Q}\left[e^{(c-1)L(\zeta)}\right] = \int_{L \leq \varepsilon} e^{(c-1)L}dQ + \int_{L > \varepsilon} e^{(c-1)L}dQ. \tag{10}$$

We next bound the integrals. For the first term, we have

$$\int_{L \leq \varepsilon} e^{(c-1)L}dQ \leq e^{(c-1)\varepsilon}.$$

While for the second integral, with the fact that $dP = e^L dQ$ and $Pr[L > \varepsilon] \leq \delta$, we can obtain

$$\int_{L > \varepsilon} e^{(c-1)L}dQ = \int_{L > \varepsilon} e^{(c-2)L}dP \leq \sup_{\zeta:L(\zeta)>\varepsilon} e^{(c-2)L(\zeta)} \cdot \delta.$$

We will now optimize $c$ in the following to conclude the proof. The $\sup_{\zeta:L(\zeta)>\varepsilon} e^{(c-2)L(\zeta)} \cdot \delta$ may be large, but for a fixed $\delta > 0$, there exists a $c > 1$ (sufficiently close to 1) such that

$$\exp[(c-1)e_{\mathcal{M}}(c)] \leq e^{(c-1)\varepsilon} + \delta \cdot H(c),$$

where $H(c)$ is a constant depending on $c$ and $\delta$. Therefore, we can rearrange the above relationship to obtain the following:

$$e_{\mathcal{M}}(c) \leq \varepsilon + \frac{\log(1 + \delta \cdot H(c))}{c - 1}.$$

Hence, for a sufficiently large $c$, i.e., $c = 1 + \sqrt{2\log(1/\delta)/\varepsilon}$, this implies that $\delta > \exp[(c-1)(e_{\mathcal{M}}(c) - \varepsilon)$, which completes the proof. □

Before the formal proof for the privacy guarantee in Theorem 1, we present another auxiliary technical lemma, which dictates the privacy amplification by uniformly sub-sampling without replacement.

**Lemma 5.** *For the mechanism induced by D2P2-SGD, $\mathcal{G}(\sigma_\epsilon) \circ \mathcal{S}(n, B)$, with $\frac{B}{n} < \frac{1}{10}$, the following privacy accountant holds:*

$$\tilde{e}(\eta) \leq \frac{7B^2\eta}{\sigma_\epsilon^2 n^2}, \ \forall \eta \leq \frac{\sigma_\epsilon^2}{2}\log\left(\frac{n}{B}\right). \tag{11}$$

*Proof.* As per-sample gradients are clipped to $l_2$ norm 1 in D2P2-SGD, this ensures the sensitivity $\Delta = 1$ for the mini-batch sum. Let $q = B/n$. Based on Lemma 3 from Abadi et al. (2016), we can obtain that

$$\tilde{e}(\eta) \leq \frac{q^2\eta}{\sigma_\epsilon^2} + \frac{3.5q^2\eta^2}{\sigma_\epsilon^4} = \frac{q^2\eta}{\sigma_\epsilon^2}\left(1 + \frac{3.5q\eta}{\sigma_\epsilon^2}\right).$$

The condition $\eta \leq \frac{\sigma_\epsilon^2}{2}\log(n/B)$ implies $\eta \leq \frac{\sigma_\epsilon^2}{2}\log(1/q)$. As $1/q \leq 1/10$, then we have:

$$\frac{3.5q\eta}{\sigma_\epsilon^2} = \frac{3.5q}{\sigma_\epsilon^2}\frac{\sigma_\epsilon^2}{2}\log(1/q) = 1.75q\log(1/q) \leq 1.75 \cdot 0.24 \approx 0.42.$$

Hence, $\tilde{e}(\eta) \leq \frac{7q^2\eta}{\sigma_\epsilon^2}$, which completes the proof. □

With this in hand, we are now ready to prove Theorem 1. We restate it in the following for completeness.

**Theorem 1:** (Privacy) Let Assumption 2 hold. There exist constants $C_1, C_2 > 0$ such that for any $\varepsilon \leq \frac{C_1 B^2 K}{n^2}$, D2P2-SGD is $(\varepsilon, \delta)$-differentially private for any $\delta > 0$, if $\sigma_\epsilon^2 \geq \frac{C_2 B^2 K^2 \ln(1/\delta)}{n^2 \varepsilon^2}$.

*Proof.* With the explicit form of noise variance we have defined in this work, i.e., $\sigma_{\epsilon,k}^2 = \frac{\sigma_\epsilon^2}{k}$, it is immediately obtained that $\sigma_{\epsilon,1}^2 > \sigma_{\epsilon,2}^2 > ... > \sigma_{\epsilon,K}^2$. If $\sigma_{\epsilon,K}^2 \geq \frac{C_2 B^2 K \ln(1/\delta)}{n^2 \varepsilon^2}$, then $\sigma_{\epsilon,k}^2 \geq \frac{C_2 B^2 K \ln(1/\delta)}{n^2 \varepsilon^2}$ for all $k$. Thus, the core of the proof has now turned to $\sigma_\epsilon^2$. As discussed before, D2P2-SGD can be treated as a composition, denoted by $\mathcal{M}$. In light of Lemma 3 and Lemma 5, it is easily obtained that

$$e_{\mathcal{M}}(\eta) \leq \frac{7K^2 B^2 \eta}{\sigma_\epsilon^2 n^2}, \; \forall \eta \leq \frac{\sigma_\epsilon^2}{2} \log\left(\frac{n}{B}\right). \tag{12}$$

$K$ is due to the $K$ iterations in D2P2-SGD and $K \leq K^2$. Additionally, based on Lemma 4, we can know that D2P2-SGD is $(\varepsilon, \delta)$-DP if there exists $\eta \leq \frac{\sigma_\epsilon^2}{2} \log\left(\frac{n}{B}\right)$ such that

$$\frac{7K^2 B^2 \eta}{\sigma_\epsilon^2 n^2} \leq \varepsilon/2, \; \exp\left(\frac{-(\eta-1)\varepsilon}{2}\right) \leq \delta. \tag{13}$$

It is now easy to verify that if $\varepsilon = C_1 B^2 K^2 / n^2$, all these conditions can be satisfied by setting $\sigma_\epsilon = \frac{C_2 BK \sqrt{\log(1/\delta)}}{n\varepsilon}$, for some explicit constants $C_1$ and $C_2$. □

## A.4 Proof for convex objectives

In this section, we will present the convergence proof for convex functions. Different from the traditional analysis to DPSGD, in D2P2-SGD, the difficulties and complexities of the proof arise from the matrix projection caused by $A_k$ and the inner product between $\mathbf{x}_k - \mathbf{x}^*$ and $\frac{\mathbf{g}_k}{\|\mathbf{g}_k\| + \gamma}$. The matrix product requires random projection properties and the relationship with the outer product of vectors. While for the inner product between $\mathbf{x}_k - \mathbf{x}^*$ and $\frac{\mathbf{g}_k}{\|\mathbf{g}_k\| + \gamma}$, we need to decompose $\mathbf{g}_k$ to $\mathbf{g}_k - \nabla f(\mathbf{x}_k)$ and $\nabla f(\mathbf{x}_k)$. Also, we have to handle the denominator $\|\mathbf{g}_k\| + \gamma$, which is a varying constant with $\|\mathbf{g}_k\|$.

We start with a well-known result regarding the convexity is presented in the following to characterize the analysis.

**Lemma 6.** *(convexity Garrigos & Gower (2023)) If $f : \mathbb{R}^d \to \mathbb{R}$ is convex and differentiable, then for $\mathbf{x}, \mathbf{y} \in \mathbb{R}^d$,*

$$f(\mathbf{x}) \geq f(\mathbf{y}) + \langle \nabla f(\mathbf{y}), \mathbf{x} - \mathbf{y} \rangle \tag{14}$$

**Theorem 2:** (Utility for convex functions) Let Assumptions 1 and 2 hold. Suppose that $f$ is a convex function and that $A$ is a random matrix with each element being sampled from a normal distribution $\mathcal{N}(0, \sigma_A^2)$. Also, let the additive noise of DP mechanism have the variance $\sigma_{\epsilon,k}^2$. If the step size $\alpha \leq \frac{1}{2L}$, then for the iterates $\{\mathbf{x}_k\}_{k=1}^K, K \geq 1$ generated by D2P2-SGD, the following relationship holds true

$$\mathbb{E}[f(\bar{\mathbf{x}}_K) - f^*] \leq \frac{\|\mathbf{x}_1 - \mathbf{x}^*\|^2 (1+\gamma)}{2\alpha K \sigma_A^2 \sqrt{p}} + \frac{\alpha p^{1.5}(1+\gamma)(\ln K + 1)\sigma_\epsilon^2}{2K} + \frac{\alpha p d^2 \sigma_A^2 (1+\gamma)}{2}, \tag{15}$$

where $\bar{\mathbf{x}}_K = \frac{1}{K} \sum_{k=1}^K \mathbf{x}_k$.

*Proof.* Let $\mathbf{g}_k^s = \nabla f(\mathbf{x}_k; s)$ and define $C_s = \frac{1}{\|\mathbf{g}_k^s\| + \gamma}$. Using the update law we have:

$$
\mathbb{E}[\|\mathbf{x}_{k+1} - \mathbf{x}^*\|^2] = \mathbb{E}[\|\mathbf{x}_k - \alpha A_k(\frac{1}{\sqrt{p}} A_k^\top \frac{1}{B} \sum_s C_s \mathbf{g}_k^s + \epsilon_k) - \mathbf{x}^*\|^2]
$$

$$
= \mathbb{E}[\|\mathbf{x}_k - \mathbf{x}^*\|^2] + \mathbb{E}[\|\alpha A_k(\frac{1}{\sqrt{p}} A_k^\top \frac{1}{B} \sum_s C_s \mathbf{g}_k^s + \epsilon_k)\|^2] - 2\alpha \mathbb{E}[\langle \frac{1}{\sqrt{p}} A_k A_k^\top \frac{1}{B} \sum_s C_s \mathbf{g}_k^s + A_k \epsilon_k, \mathbf{x}_k - \mathbf{x}^* \rangle]
$$

$$
= \mathbb{E}[\|\mathbf{x}_k - \mathbf{x}^*\|^2] + \mathbb{E}[\|\alpha(\frac{1}{\sqrt{p}} A_k A_k^\top \frac{1}{B} \sum_s C_s \mathbf{g}_k^s + A_k \epsilon_k)\|^2] - 2\alpha \mathbb{E}[\langle \frac{1}{\sqrt{p}} \mathbb{E}[A_k A_k^\top] \frac{\mathbf{g}_k}{\|\mathbf{g}_k\| + \gamma}, \mathbf{x}_k - \mathbf{x}^* \rangle]
$$

$$
= \mathbb{E}[\|\mathbf{x}_k - \mathbf{x}^*\|^2] + \mathbb{E}[\|\alpha(\frac{1}{\sqrt{p}} A_k A_k^\top \frac{1}{B} \sum_s C_s \mathbf{g}_k^s + A_k \epsilon_k)\|^2] - 2\alpha \mathbb{E}[\langle \frac{1}{\sqrt{p}} p\sigma_A^2 I \frac{\mathbf{g}_k}{\|\mathbf{g}_k\| + \gamma}, \mathbf{x}_k - \mathbf{x}^* \rangle]
$$

$$
= \mathbb{E}[\|\mathbf{x}_k - \mathbf{x}^*\|^2] + \mathbb{E}[\|\alpha(\frac{1}{\sqrt{p}} A_k A_k^\top \frac{1}{B} \sum_s C_s \mathbf{g}_k^s + A_k \epsilon_k)\|^2] - 2\alpha \sqrt{p} \sigma_A^2 \mathbb{E}[\langle \mathbf{x}_k - \mathbf{x}^*, \frac{\mathbf{g}_k}{\|\mathbf{g}_k\| + \gamma} \rangle]
$$

$$
= \mathbb{E}[\|\mathbf{x}_k - \mathbf{x}^*\|^2] + \alpha^2 \mathbb{E}[\|\frac{1}{\sqrt{p}} A_k A_k^\top \frac{1}{B} \sum_s C_s \mathbf{g}_k^s\|^2] + \alpha^2 \mathbb{E}[\|A_k \epsilon_k\|^2] - 2\alpha \sqrt{p} \sigma_A^2 \mathbb{E}[\langle \mathbf{x}_k - \mathbf{x}^*, \frac{\mathbf{g}_k}{\|\mathbf{g}_k\| + \gamma} \rangle]
$$

$$(16)$$

The third equality follows from that $\mathbb{E}[A_k \epsilon_k] = 0$ based on the Concentration Theorem for Projections Dasgupta et al. (2012), the independence between $A_k A_k^\top$ and $\mathbf{g}(\mathbf{x}_k)$, and $\mathbb{E}[\frac{1}{B} \sum_s C_s \mathbf{g}_k^s] = \frac{\mathbf{g}_k}{\|\mathbf{g}_k\| + \gamma}$. According to Lemma 2, we know that $\mathbb{E}[A_k A_k^\top] = p\sigma_A^2 I$ Nabil (2017), which yields the fourth equality. The last equality follows similarly from $\mathbb{E}[A_k \epsilon_k] = 0$. As $\mathbb{E}[\|A_k A_k^\top\|^2] = \mathbb{E}[\|\sum_{i=1}^p A_{k,i} A_{k,i}^\top\|^2] \le p \sum_{i=1}^p \mathbb{E}[\|A_{k,i} A_{k,i}^\top\|^2] \le p \sum_{i=1}^p \mathbb{E}[\|A_{k,i}\|^2] \mathbb{E}[\|A_{k,i}^\top\|^2] = p^2 d^2 \sigma_A^4$. Also, $\mathbb{E}[\|A_k \epsilon_k\|^2] = \mathbb{E}[\epsilon_k^\top A_k^\top A_k \epsilon_k] = \mathbb{E}[\epsilon_k^\top \mathbb{E}[A_k^\top A_k] \epsilon_k] = \epsilon_k^\top p\sigma_A^2 I \epsilon_k = \mathbb{E}[p\sigma_A^2 \|\epsilon_k\|^2]$. With $\|C_s \mathbf{g}_k^s\| \le 1$, the following relationship holds:

$$
\mathbb{E}[\|\mathbf{x}_{k+1} - \mathbf{x}^*\|^2] \le \mathbb{E}[\|\mathbf{x}_k - \mathbf{x}^*\|^2] + \alpha^2 p^{1.5} d^2 \sigma_A^4 + \alpha^2 p^2 \sigma_A^2 \sigma_{\epsilon,k}^2 - 2\alpha \sqrt{p} \sigma_A^2 \mathbb{E}[\langle \mathbf{x}_k - \mathbf{x}^*, \frac{\mathbf{g}_k - \nabla f(\mathbf{x}_k) + \nabla f(\mathbf{x}_k)}{\|\mathbf{g}_k\| + \gamma} \rangle]
$$

$$
\le \mathbb{E}[\|\mathbf{x}_k - \mathbf{x}^*\|^2] + \alpha^2 p^{1.5} d^2 \sigma_A^4 + \alpha^2 p^2 \sigma_A^2 \sigma_{\epsilon,k}^2 + 2\alpha \sqrt{p} \sigma_A^2 \mathbb{E}[\langle \mathbf{x}^* - \mathbf{x}_k, \frac{\nabla f(\mathbf{x}_k)}{\|\mathbf{g}_k\| + \gamma} \rangle]
$$

$$
+ 2\alpha \sqrt{p} \sigma_A^2 \mathbb{E}[\langle \mathbf{x}^* - \mathbf{x}_k, \frac{\mathbf{g}_k - \nabla f(\mathbf{x}_k)}{\|\mathbf{g}_k\| + \gamma} \rangle]
$$

$$(17)$$

As $\mathbb{E}[\mathbf{g}_k] = \nabla f(\mathbf{x}_k)$, the last term on the right hand side of the above inequality equals 0. Using Lemma 6 and Cauchy-Schwartz inequality yields the following,

$$
\mathbb{E}[\|\mathbf{x}_{k+1} - \mathbf{x}^*\|^2] \le \mathbb{E}[\|\mathbf{x}_k - \mathbf{x}^*\|^2] + \alpha^2 p^{1.5} d^2 \sigma_A^4 + \alpha^2 p^2 \sigma_A^2 \sigma_{\epsilon,k}^2 + \frac{2\alpha \sigma_A^2 \sqrt{p}}{\|\mathbf{g}_k\| + \gamma} \mathbb{E}[f^* - f(\mathbf{x}_k)] \tag{18}
$$

Then, we can obtain

$$
\frac{2\alpha \sigma_A^2 \sqrt{p}}{\|\mathbf{g}_k\| + \gamma} \mathbb{E}[f(\mathbf{x}_k) - f^*] \le \mathbb{E}[\|\mathbf{x}_k - \mathbf{x}^*\|^2] + \alpha^2 p^{1.5} d^2 \sigma_A^4 + \alpha^2 p^2 \sigma_A^2 \sigma_{\epsilon,k}^2 - \mathbb{E}[\|\mathbf{x}_{k+1} - \mathbf{x}^*\|^2] \tag{19}
$$

Based on the definition of the clipping mechanism, we know that $\mathbf{g} \leftarrow \mathbf{g} \cdot \frac{1}{\|\mathbf{g}\| + \gamma}$ ensures $\|\mathbf{g}_k\| \le 1$ for all $k > 0$. Therefore, diving both sides of the last inequality by $\frac{2\alpha \sigma_A^2 \sqrt{p}}{\|\mathbf{g}_k\| + \gamma}$ and applying $\|\mathbf{g}_k\| \le 1$ produces the following

$$
\mathbb{E}[f(\mathbf{x}_k) - f^*] \le \frac{\mathbb{E}[\|\mathbf{x}_k - \mathbf{x}^*\|^2 - \|\mathbf{x}_{k+1} - \mathbf{x}^*\|^2]}{2\sqrt{p} \sigma_A^2 \alpha}(1 + \gamma) + \frac{\alpha p d^2 \sigma_A^2 (1 + \gamma)}{2} + \frac{\alpha p^{1.5} \sigma_{\epsilon,k}^2 (1 + \gamma)}{2}
$$

$$
\le \frac{\mathbb{E}[\|\mathbf{x}_k - \mathbf{x}^*\|^2 - \|\mathbf{x}_{k+1} - \mathbf{x}^*\|^2]}{2\sqrt{p} \sigma_A^2 \alpha}(1 + \gamma) + \frac{\alpha p d^2 \sigma_A^2 (1 + \gamma)}{2} + \frac{\alpha p^{1.5} \sigma_{\epsilon,k}^2 (1 + \gamma)}{2}
$$

$$(20)$$

Summing over $k$ from 1 to $K$ and dividing both sides by $K$ produces the following relationship:

$$\frac{1}{K}\sum_{k=1}^{K}\mathbb{E}[f(\mathbf{x}_k)-f^*] \le \frac{\mathbb{E}[\|\mathbf{x}_1-\mathbf{x}^*\|^2}{2K\alpha\sqrt{p}\sigma_A^2}(1+\gamma) + \frac{\alpha pd^2\sigma_A^2(1+\gamma)}{2} + \frac{\alpha p^{1.5}(1+\gamma)}{2K}\sum_{k=1}^{K}\sigma_{\epsilon,k}^2 \tag{21}$$

Using Jensen's inequality on $f(\mathbf{x}_k)-f^*$ (i.e., $f(\bar{\mathbf{x}}_K)-f^* \le \frac{1}{K}\sum_{k=1}^{K}\mathbb{E}[f(\mathbf{x}_k)-f^*]$) completes the proof. $\square$

**Corollary 1.** With conditions defined in Theorem 2, when $\alpha = \mathcal{O}(\frac{1}{\sqrt{K}})$, the following relationship holds true, i.e., $\mathbb{E}[f(\bar{\mathbf{x}}_K)-f^*] \le \mathcal{O}(\frac{1}{\sqrt{K}} + \frac{\ln K}{K^{1.5}})$.

*Proof.* Based on the conclusion from Theorem 2, substituting $\alpha = \mathcal{O}(\frac{1}{\sqrt{K}})$ into it and applying the upper bound of the partial sum leads to the desirable result. $\square$

### A.5  Impact of Privacy and Dimensions for Convex Functions

In this subsection, we derive the explicit impact of the privacy budget $\varepsilon$ and dimensions, $d$ and $p$ on the error bound and compare it to that in DPSGD. Recall the conclusion from Theorem 2 in the following:

$$\mathbb{E}[f(\bar{\mathbf{x}}_K)-f^*] \le \frac{\|\mathbf{x}_1-\mathbf{x}^*\|^2(1+\gamma)}{2\alpha K\sigma_A^2\sqrt{p}} + \frac{\alpha p^{1.5}(1+\gamma)(\ln K+1)\sigma_\epsilon^2}{2K} + \frac{\alpha pd^2\sigma_A^2(1+\gamma)}{2} \tag{22}$$

Let $D = \|\mathbf{x}_1-\mathbf{x}^*\|^2$ and set $\gamma \approx 0$ for simplicity such that the last inequality can be rewritten as

$$\mathbb{E}[f(\bar{\mathbf{x}}_K)-f^*] \le \mathcal{O}\left(\frac{D}{\alpha K\sigma_A^2\sqrt{p}} + \frac{\alpha p^{3/2}\ln K\sigma_\epsilon^2}{K} + \alpha pd^2\sigma_A^2\right). \tag{23}$$

Let $\sigma_\epsilon^2 = \frac{C_2 B^2 K^2\ln(1/\delta)}{n^2\epsilon^2}$, $C_2 = \frac{n\varepsilon}{p^{5/2}K\ln K\sqrt{\ln(1/\delta)}}$ and $B = \mathcal{O}(1)$. Also, based on Corollary 1, we set step size $\alpha = \frac{p^{3/2}}{d^2\sqrt{K}}$. Substituting $\sigma_\epsilon^2, C_2, B, \alpha$ into the last inequality, we have

$$\mathbb{E}[f(\bar{\mathbf{x}}_K)-f^*] \le \mathcal{O}\left(\frac{1}{\sqrt{K}}\cdot\left(\underbrace{\frac{Dd^2}{p^2\sigma_A^2}}_{\text{Initialization error}} + \underbrace{\frac{\sqrt{p\ln(1/\delta)}}{d^2n\varepsilon}}_{\text{Privacy error}} + \underbrace{p^{5/2}\sigma_A^2}_{\text{Projection error}}\right)\right). \tag{24}$$

The last inequality states that the error bound is impacted by initialization error, privacy error, and projection error. Recalling from Bassily et al. (2014), we know that the vanilla DPSGD has achieved the optimal convergence rate of $\mathcal{O}\left(\frac{LD}{\sqrt{K}}\left(1+\frac{\sqrt{d\ln(1/\delta)}}{n\varepsilon}\right)\right)$. Comparing the privacy errors in both D2P2-SGD and DPSGD yields that the former has reduced the negative impact of privacy error significantly due to the random projection, but at the cost of additional approximation error $p^{5/2}\sigma_A^2$ caused by the random projection. Additionally, the initialization error in D2P2-SGD is also impacted by the dimensions $d$ and $p$. With random projection, the tradeoff between utility and privacy in D2P2-SGD becomes more complex compared to DPSGD. This also requires a careful selection of reduced dimension $p$ and a proper sampling distribution for the projection matrix $A$ in D2P2-SGD to achieve better performance.

### A.6  Proof for non-convex objectives

In this subsection, we present the missing proof for the non-convex objectives. Unlike the convex setting, our non-convex analysis faces distinct challenges arising from the interaction between the inner product of batch gradient and stochastic gradient, $\langle\nabla f(\mathbf{x}_k), \mathbf{g}_k\rangle$ and the random projection matrix product. To resolve these, we must simultaneously leverage: (i) the independence between random projections and mini-batch sampling, and (ii) triangle inequality decomposition of the gradient norms $\|\nabla f(\mathbf{x}_k)\|$ and $\|\mathbf{g}_k\|$.

**Theorem 3:** (Utility for non-convex functions) Let Assumptions 1 and 2 hold. Suppose that $A$ is a random matrix with each element being sampled from a normal distribution $\mathcal{N}(0, \sigma_A^2)$. Also, let the additive noise of

the DP mechanism have the variance $\sigma^2_{\epsilon,k}$. If the step size $\alpha \leq \frac{1}{2L}$, and, then for the iterates $\{\mathbf{x}_k\}^K_{k=1}, K \geq 1$ generated by D2P2-SGD, the following relationship holds true

$$\min_{k\in[1,K]}\mathbb{E}[\|\nabla f(\mathbf{x}_k)\|] \leq \frac{f(\mathbf{x}_1) - f^*}{K\sigma^2_A\sqrt{p}\alpha} + \frac{\alpha L p^{1.5}\sigma^2_\epsilon(\ln K + 1)}{K} + L\alpha\sqrt{p}d^2\sigma^2_A + 2\sigma + \gamma. \tag{25}$$

*Proof.* Due to the smoothness condition, we have the following relationship:

$$f(\mathbf{x}_{k+1}) \leq f(\mathbf{x}_k) + \langle \nabla f(\mathbf{x}_k), \mathbf{x}_{k+1} - \mathbf{x}_k \rangle + \frac{L}{2}\|\mathbf{x}_{k+1} - \mathbf{x}_k\|^2 \tag{26}$$

Also, we know that $\mathbf{x}_{k+1} - \mathbf{x}_k = -\alpha A_k(\frac{1}{\sqrt{p}}A_k^\top \frac{1}{B}\sum_s C_s\mathbf{g}_k^s + \epsilon_k)$.

Substituting the above equality into the smoothness equation yields the following relationship:

$$f(\mathbf{x}_{k+1}) \leq f(\mathbf{x}_k) + \langle \nabla f(\mathbf{x}_k), -\alpha A_k(\frac{1}{\sqrt{p}}A_k^\top \frac{1}{B}\sum_s C_s\mathbf{g}_k^s + \epsilon_k)\rangle$$
$$+ \frac{L\alpha^2}{2}\|A_k(\frac{1}{\sqrt{p}}A_k^\top \frac{1}{B}\sum_s C_s\mathbf{g}_k^s + \epsilon_k)\|^2 \tag{27}$$

Taking expectation on both sides, with the proof from Theorem 2, we can obtain the following relationships

$$\mathbb{E}[f(\mathbf{x}_{k+1})] \leq \mathbb{E}[f(\mathbf{x}_k)] - \alpha\mathbb{E}\left[\langle \nabla f(\mathbf{x}_k), \frac{1}{\sqrt{p}}A_kA_k^\top \frac{1}{B}\sum_s C_s\mathbf{g}_k^s + A_k\epsilon_k\rangle\right]$$

$$+ \frac{L\alpha^2}{2}\mathbb{E}\left[\|\frac{1}{\sqrt{p}}A_kA_k^\top \frac{1}{B}\sum_s C_s\mathbf{g}_k^s + A_k\epsilon_k\|^2\right]$$

$$\leq \mathbb{E}[f(\mathbf{x}_k)] - \alpha\mathbb{E}[\langle \nabla f(\mathbf{x}_k), \frac{1}{\sqrt{p}}A_kA_k^\top \frac{1}{B}\sum_s C_s\mathbf{g}_k^s\rangle] + \frac{L\alpha^2}{2}\mathbb{E}\left[\|\frac{1}{\sqrt{p}}A_kA_k^\top \frac{1}{B}\sum_s C_s\mathbf{g}_k^s\|^2 + \|A_k\epsilon_k\|^2\right]$$

$$= \mathbb{E}[f(\mathbf{x}_k)] - \alpha\mathbb{E}[\langle \nabla f(\mathbf{x}_k), \frac{1}{\sqrt{p}}A_kA_k^\top \frac{1}{B}\sum_s C_s\mathbf{g}_k^s\rangle] + \frac{L\alpha^2}{2}\mathbb{E}\left[\|\frac{1}{\sqrt{p}}A_kA_k^\top \frac{1}{B}\sum_s C_s\mathbf{g}_k^s\|^2\right] + \frac{L\alpha^2}{2}\mathbb{E}\left[\|A_k\epsilon_k\|^2\right]$$

$$= \mathbb{E}[f(\mathbf{x}_k)] - \alpha\mathbb{E}[\langle \nabla f(\mathbf{x}_k), \frac{1}{\sqrt{p}}\mathbb{E}[A_kA_k^\top]\frac{\mathbf{g}_k}{\|\mathbf{g}_k\| + \gamma}\rangle] + \frac{L\alpha^2}{2}\mathbb{E}\left[\|\frac{1}{\sqrt{p}}A_kA_k^\top \frac{1}{B}\sum_s C_s\mathbf{g}_k^s\|^2\right] + \frac{L\alpha^2}{2}\mathbb{E}\left[\|A_k\epsilon_k\|^2\right] \tag{28}$$

The second inequality is due to the expectation of $A_k\epsilon_k$ equal to 0, as shown in the generally convex case. From random projection property we know $\mathbb{E}[A_kA_k^\top] = p\sigma^2_A I$, so we will have:

$$\mathbb{E}[f(\mathbf{x}_{k+1})] \leq \mathbb{E}[f(\mathbf{x}_k)] - \alpha\sqrt{p}\sigma^2_A\mathbb{E}[\langle \nabla f(\mathbf{x}_k) - \mathbf{g}_k + \mathbf{g}_k, \frac{\mathbf{g}_k}{\|\mathbf{g}_k\| + \gamma}\rangle]$$

$$+ \frac{L\alpha^2}{2}\mathbb{E}\left[\|\frac{1}{\sqrt{p}}A_kA_k^\top \frac{1}{B}\sum_s C_s\mathbf{g}_k^s\|^2\right] + \frac{L\alpha^2}{2}\mathbb{E}\left[\|A_k\epsilon_k\|^2\right] \tag{29}$$

As $-\langle \mathbf{a}, \mathbf{b}\rangle \leq \|\mathbf{a}\|\|\mathbf{b}\|$ and $\mathbf{g}_k^\top\mathbf{g}_k = \|\mathbf{g}_k\|^2$, we have:

$$\mathbb{E}[f(\mathbf{x}_{k+1})] \leq \mathbb{E}[f(\mathbf{x}_k)] + \alpha\sqrt{p}\sigma^2_A\mathbb{E}[\|\nabla f(\mathbf{x}_k) - \mathbf{g}_k\|\frac{\|\mathbf{g}_k\|}{\|\mathbf{g}_k\| + \gamma}]\frac{L\alpha^2}{2}pd^2\sigma^4_A + \frac{L\alpha^2}{2}p^2\sigma^2_A\sigma^2_{\epsilon,k} - \alpha\sqrt{p}\sigma^2_A\mathbb{E}[\frac{\|\mathbf{g}_k\| + \gamma}{\|\mathbf{g}_k\| + \gamma}\|\mathbf{g}_k\|]$$

$$+ \alpha\sqrt{p}\sigma^2_A\mathbb{E}[\frac{\gamma}{\|\mathbf{g}_k\| + \gamma}\|\mathbf{g}_k\|] \tag{30}$$

Since $\frac{\|\mathbf{g}_k\|}{\|\mathbf{g}_k\|+\gamma} \leq 1$, we have $\alpha\sqrt{p}\sigma^2_A\mathbb{E}[\|\nabla f(\mathbf{x}_k) - \mathbf{g}_k\|\frac{\|\mathbf{g}_k\|}{\|\mathbf{g}_k\|+\gamma}] \leq \alpha\sqrt{p}\sigma^2_A\mathbb{E}[\|\nabla f(\mathbf{x}_k) - \mathbf{g}_k\|]$. Hence, with $\|\nabla f(\mathbf{x}_k)\| \leq \|\nabla f(\mathbf{x}_k) - \mathbf{g}_k\| + \|\mathbf{g}_k\|$ the following relationship can be obtained:

$$\alpha\sqrt{p}\sigma^2_A\mathbb{E}[\|\nabla f(\mathbf{x}_k)\|] \leq \mathbb{E}[f(\mathbf{x}_k) - f(\mathbf{x}_{k+1})] + 2\alpha\sqrt{p}\sigma^2_A\mathbb{E}[\|\nabla f(\mathbf{x}_k) - \mathbf{g}_k\|]$$

$$+ \frac{L\alpha^2}{2}pd^2\sigma^4_A + \frac{L\alpha^2}{2}p^2\sigma^2_A\sigma^2_{\epsilon,k} + \alpha\sqrt{p}\sigma^2_A\mathbb{E}[\frac{\gamma}{\|\mathbf{g}_k\| + \gamma}\|\mathbf{g}_k\|] \tag{31}$$

Dividing both sides of the last inequality by $\alpha\sqrt{p}\sigma_A^2$ yields:

$$
\mathbb{E}[\|\nabla f(\mathbf{x}_k)\|] \leq \frac{\mathbb{E}[f(\mathbf{x}_k) - f(\mathbf{x}_{k+1})]}{\alpha\sqrt{p}\sigma_A^2} + 2\mathbb{E}[\|\nabla f(\mathbf{x}_k) - \mathbf{g}_k\|]
$$
$$
+ \frac{L\alpha\sqrt{p}d^2\sigma_A^2}{2} + \frac{L\alpha p^{1.5}\sigma_{\epsilon,k}^2}{2} + \gamma
\tag{32}
$$

which is due to $\frac{\gamma}{\|\mathbf{g}_k\|+\gamma}\|\mathbf{g}_k\| \leq \gamma$. Since $\mathbb{E}[\|\nabla f(\mathbf{x}_k) - \mathbf{g}_k\|] \leq \sqrt{\mathbb{E}[\|\nabla f(\mathbf{x}_k) - \mathbf{g}_k\|^2]} \leq \frac{\sigma}{\sqrt{B}}$ Summing the last equation over from 1 to $K$ and dividing both sides by $K$ grants us the following relationship:

$$
\frac{1}{K}\sum_{k=1}^{K}\mathbb{E}[\|\nabla f(\mathbf{x}_k)\|] \leq \frac{f(\mathbf{x}_1)}{\alpha\sigma_A^2\sqrt{p}K} + 2\frac{\sigma}{\sqrt{B}} + \gamma + \frac{L\alpha\sqrt{p}d^2\sigma_A^2}{2} + \frac{L\alpha p^{1.5}}{2K}\sum_{k=1}^{K}\sigma_{\epsilon,k}^2
\tag{33}
$$

With the fact that $\sum_{k=1}^{K}\sigma_{\epsilon,k}^2 \leq (\ln K + 1)\sigma_\epsilon^2$ and that the minimum is less than the average, the desirable result is obtained. $\square$

**Corollary 2.** With conditions defined in Theorem 3, when $\alpha = \mathcal{O}(\frac{1}{\sqrt{K}})$, the following relationship hold true, $\min_{k\in[1,K]}\mathbb{E}[\|\nabla f(\mathbf{x}_k)\|] \leq \mathcal{O}(\frac{1}{\sqrt{K}} + \frac{\ln K}{K^{1.5}} + \sigma + \gamma)$.

*Proof.* Using the conclusion from Theorem 3 and substituting the step size $\alpha$ into the conclusion yields the desirable result. $\square$

### A.7 Comparison of the Clipping Bias

In this subsection, we show the comparison among different methods for the clipping bias in Table 4. It suggests that our clipping bias is proportional to $\sigma$, which is the same as in existing works. However, due to the adoption of per-sample gradient normalization, the stability constant $\gamma$ also affects the clipping bias. In practice, $\gamma$ is a small positive constant similar to the one in the Adam optimizer.

Table 4: Comparison among different methods.

| Method | Clipping Bias |
|---|---|
| Chen et al. (2020) | Wasserstein distance |
| Koloskova et al. (2023a) | $\sigma$ or $\sigma^2/a$ |
| Xiao et al. (2023) | $15\sigma$ |
| Bu et al. (2024) | $\sigma/r$ |
| D2P2-SGD (Ours) | $2\sigma/\sqrt{B} + \gamma$ |

$a > 0$ is the clipping threshold; $r > 0$

### A.8 Impact of Privacy and Dimensions for Non-convex Functions

In this subsection, we derive the explicit impact of the privacy budget $\varepsilon$ and dimensions, $d$ and $p$ on the error bound. Recall the conclusion from Theorem 3 in the following:

$$
\min_{k\in[1,K]}\mathbb{E}[\|\nabla f(\mathbf{x}_k)\|] \leq \frac{f(\mathbf{x}_1) - f^*}{K\sigma_A^2\sqrt{p}\alpha} + \frac{\alpha L p^{1.5}\sigma_\epsilon^2(\ln K + 1)}{K} + L\alpha\sqrt{p}d^2\sigma_A^2 + \frac{2\sigma}{\sqrt{B}} + \gamma.
\tag{34}
$$

Let $D = f(\mathbf{x}_1) - f^*$ such that the last inequality can be rewritten as

$$
\mathbb{E}[f(\bar{\mathbf{x}}_K) - f^*] \leq \mathcal{O}\left(\frac{D}{\alpha K\sigma_A^2\sqrt{p}} + \frac{\alpha p^{3/2}\ln K L\sigma_\epsilon^2}{K} + L\alpha\sqrt{p}d^2\sigma_A^2 + \frac{\sigma}{\sqrt{B}} + \gamma\right).
\tag{35}
$$

Published in Transactions on Machine Learning Research (09/2025)

Let $\sigma_\epsilon^2 = \frac{C_2 B^2 K^2 \ln(1/\delta)}{n^2 \epsilon^2}$, $C_2 = \frac{n\varepsilon}{p^{5/2} K \ln K \sqrt{\ln(1/\delta)} L}$ and $B = \mathcal{O}(1)$. Also, based on Corollary 2, we set step size $\alpha = \frac{p^{3/2}}{d^2 \sqrt{K}}$. Substituting $\sigma_\epsilon^2, C_2, B, \alpha$ into the last inequality, we have

$$\mathbb{E}[f(\bar{\mathbf{x}}_K) - f^*] \leq \mathcal{O}\left(\frac{1}{\sqrt{K}} \cdot \left(\underbrace{\frac{Dd^2}{p^2 \sigma_A^2}}_{\text{Initialization error}} + \underbrace{\frac{\sqrt{p\ln(1/\delta)}}{d^2 n\varepsilon}}_{\text{Privacy error}} + \underbrace{p^2 \sigma_A^2}_{\text{Projection error}}\right) + \underbrace{\sigma + \gamma}_{\text{Clipping bias}}\right). \quad (36)$$

The last inequality reveals that the error bound is dictated by initialization error, privacy error, projection error, and clipping bias. Different from convex functions, the asymptotic convergence for non-convex functions leads to the clipping bias, which worsens the utility when $K$ is sufficiently large. When $K$ is small, corresponding to the early phase of optimization, the tradeoff between utility and privacy dominates. Analogously, this requires a meticulous selection of dimension $p$ and a proper sampling distribution for the projection matrix in D2P2-SGD to achieve better performance if non-asymptotic convergence is favored. If $\gamma$ is set $\mathcal{O}(\frac{1}{\sqrt{K}})$, the clipping bias is $\sigma$, which resembles the result in Koloskova et al. (2023a).

## A.9 Additional Results

All the experiments were conducted on a machine equipped with an Intel Xeon Silver 4110 CPU and an NVIDIA Titan RTX GPU.

### A.9.1 Additional Datasets

Figure 7 shows the performance with ResNet20 Wang et al. (2019a) on the CIFAR-10 dataset. From the plots, we can observe that D2P-SGD slightly performs better than D2P2-SGD due to the random projection error in the error bound, which also deteriorates the performance of DP2-SGD. The dynamic variance mechanism plays a central role in enhancing performance closer to SGD. However, the tradeoff between random projection and privacy drives the performance of D2P2-SGD between those in D2P-SGD (upper) and DP2-SGD (lower).

In Figures 8 and 9, results for the CIFAR-10 dataset are provided, showing a similar trend to the other datasets. We also compare them to disclose the impact of the reduction rate. The reduction rate is the number of dimensions that have been compressed in random projection. For example, if $d = 10000$ and the reduction rate is 0.3, then $p = 7000$. The chosen hyperparameters are: batch size = 1024, $\sigma_\epsilon = 2.0$, and dimension reduction rate = 0.3 in Figure 8. If we decrease the dimension reduction rate to 0.1 in Figure 9, we can observe that the performance is enhanced, but the privacy loss remains the same, which aligns with our findings in the main contents.

In Figure 10, when reducing $\sigma_\epsilon = 1.0$, we can see a significant accuracy improvement, but with a sacrifice of privacy loss instead. However, D2P2-SGD achieves the best performance over baselines and is favorably comparable to SGD. Similarly, for Figures 11-13 (KMNIST, EMNIST, MNIST), D2P2-SGD is favorably comparable to or outperforms all baselines, which strengthens our claims.

### A.9.2 Calculation of $C_1$ value

From Figures 2 and 3, given the batch size equal to 1024, the number of epoch 40, and the training data sizes 60000 and 73257, based on the upper bound for $\varepsilon$ in Theorem 1, $\varepsilon \leq \frac{C_1 B^2 K}{n^2}$, we can obtain that as long as $C_1 \geq 314$, $\varepsilon$ values for both datasets will remain within the bound.

### A.9.3 Detail of network architecture

### A.9.4 Hyperparameter Setup

In this section, we detail the key hyperparameters we have used in this work. All hyperparameters are subject to manual tuning, though a hyperparameter optimization method is likely beneficial for potential performance improvements.

24

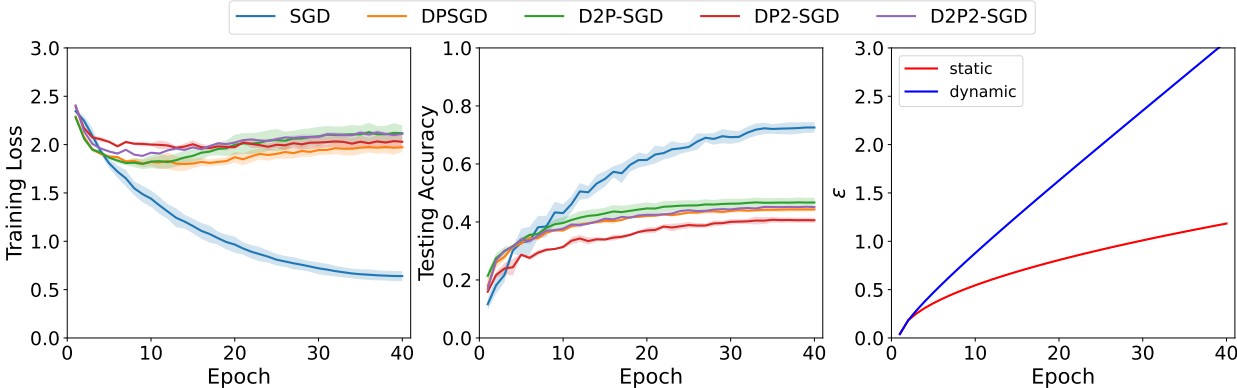

Figure 7: Comparison among different methods for CIFAR10 data with ResNet20: on the right side, the privacy loss is shown for static and dynamic scenarios.

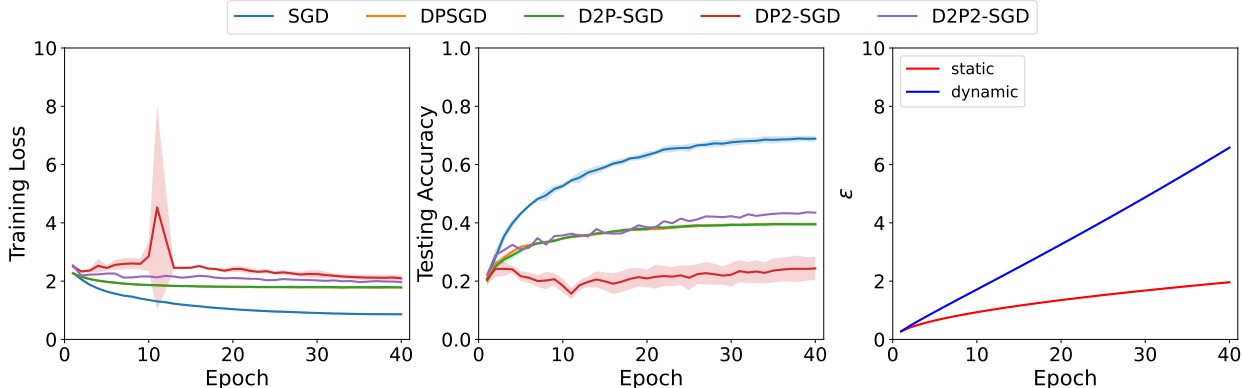

Figure 8: Comparison among different methods for CIFAR10 data with reduction rate being 0.3, with CNN: on the right side, the privacy loss is shown for static and dynamic scenarios.

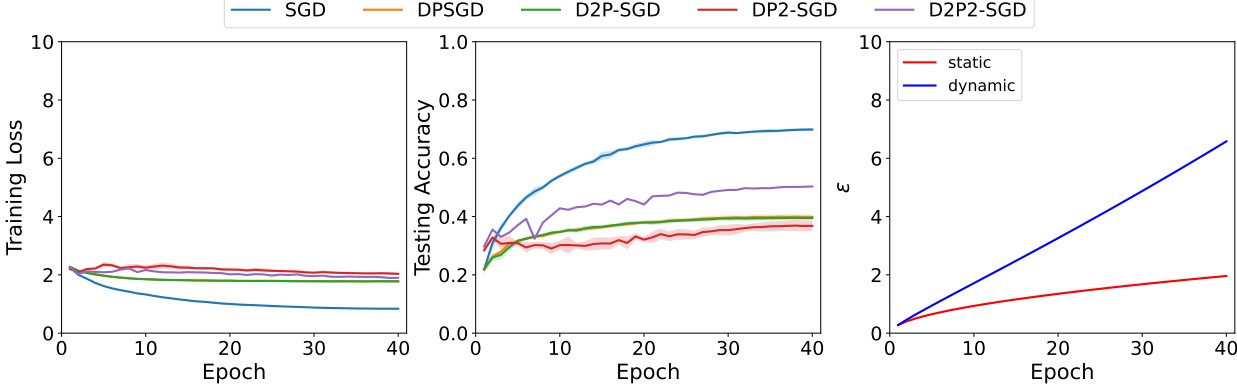

Figure 9: Comparison among different methods for CIFAR10 data with reduction rate being 0.1, with CNN: on the right side, the privacy loss is shown for static and dynamic scenarios.

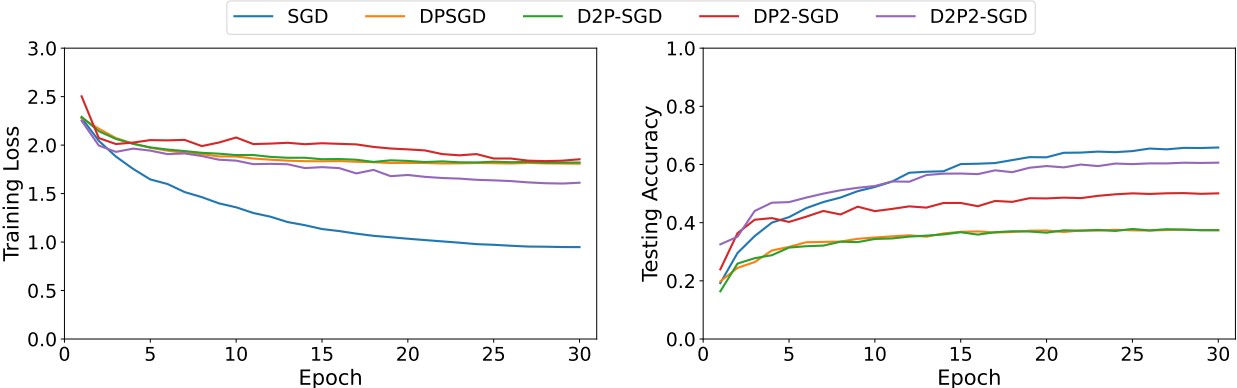

Figure 10: Comparison among different methods for CIFAR10 data with CNN: training loss and testing accuracy.

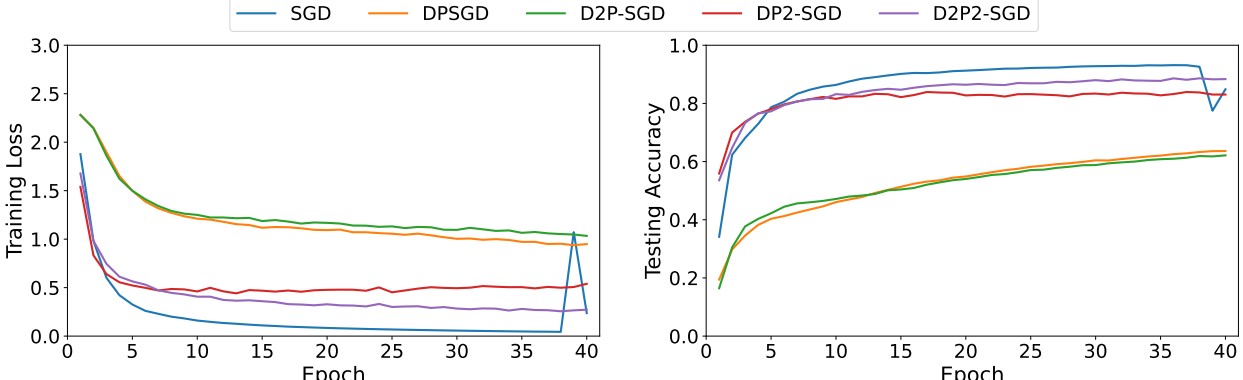

Figure 11: Comparison among different methods for KMNIST data with CNN: training loss and testing accuracy.

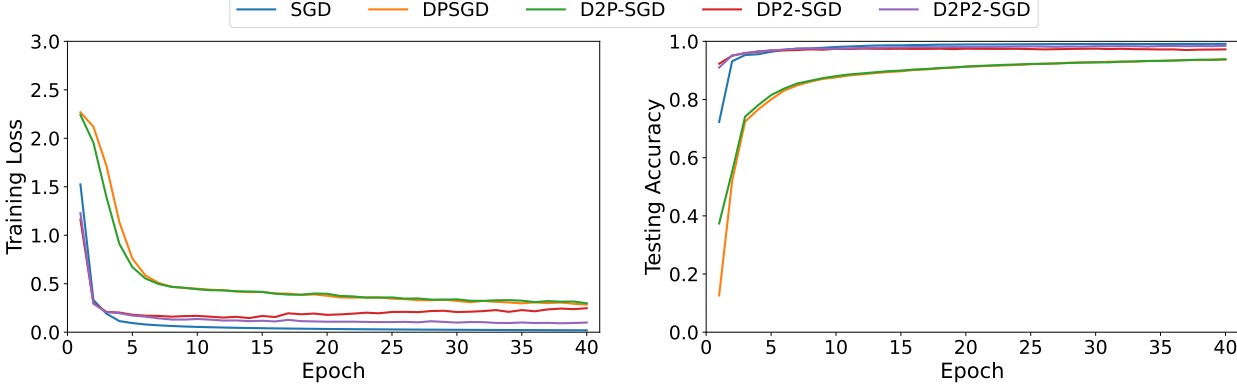

Figure 12: Comparison among different methods for EMNIST data with CNN: training loss and testing accuracy.

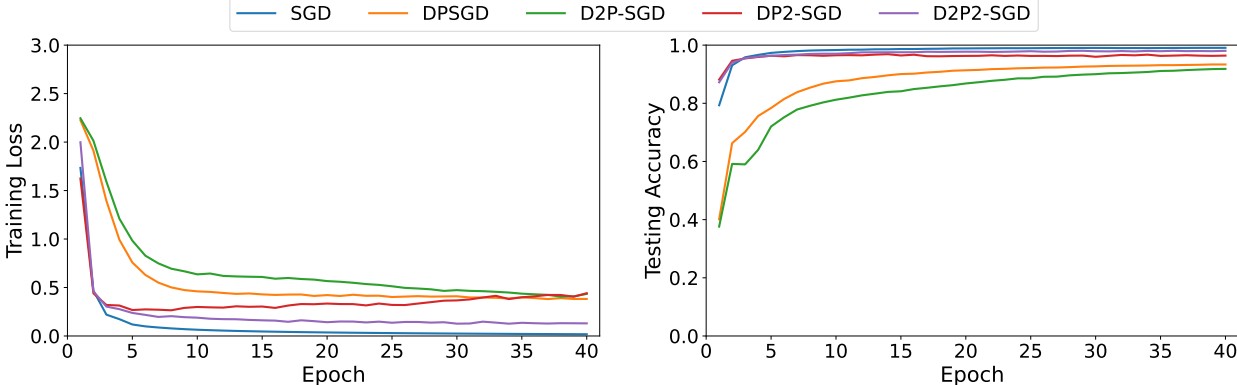

Figure 13: Comparison among different methods for MNIST data with CNN: training loss and testing accuracy.

Table 5: Network architecture for FashionMNIST, SVHN, and CIFAR-10 datasets.

| Layer | Parameters |
|---|---|
| Convolution | 16 filters of $3 \times 3$, strides 1 |
| Average Pooling | $2 \times 2$ |
| Convolution | 32 filters of $3 \times 3$, strides 1 |
| Average Pooling | $2 \times 2$ |
| Convolution | 32 filters of $3 \times 3$, strides 1 |
| Average Pooling | $2 \times 2$ |
| Convolution | 64 filters of $3 \times 3$, strides 1 |
| Adaptive Average Pooling | $1 \times 1$ |
| Fully connected | 64 units |
| Softmax | 10 units |

Table 6: Hyperparameters for experiments.

| Hyperparameter | Value |
|---|---|
| Learning rate $\alpha$ | 0.01 |
| Clipping parameter $\gamma$ | 0.01 |
| Batch size $B$ | (256, 512, 1024) |
| Number of Epoch $K$ | 40 |
| Injected noise variance $\sigma_\epsilon$ | 3.0 |
| Sampling variance | 1 |
| Percentage of dimensionality reduction | 0.7 |
| Number of random seeds | 4 |

