# OpenReview forum: "Balancing Utility and Privacy: Dynamically Private SGD with Random Projection"
_TMLR — Accepted by TMLR_

### Review · Reviewer_JbaZ · 2025-06-13

**Summary Of Contributions:**

The submission introduces D2P2-SGD, a novel differentially private stochastic gradient descent optimizer that integrates dynamic differential privacy (DDP) with automatic gradient clipping and random projection. This approach aims to balance privacy, utility, and computational complexity in large-scale deep learning models. D2P2-SGD employs a dynamic noise mechanism with time-varying variance and random projection to reduce noise dimensionality, coupled with per-sample gradient normalization for automatic clipping. The authors prove sub-linear convergence rates for both convex and non-convex objectives, matching standard SGD rates. Experiments on datasets like FashionMNIST, SVHN, and CIFAR-10 demonstrate improved accuracy over baselines (DPSGD, D2P-SGD, DP2-SGD) while maintaining differential privacy, though with increased privacy loss due to dynamic noise.

**Audience:**

Yes

**Claims And Evidence:**

No

**Requested Changes:**

1. Improved Proof of Theorem 1 (Critical): Please, provide a complete proof of Theorem 1 with full bibliographical references for Lemma 3. Explicitly address the stochasticity of random projections in the privacy analysis to clarify its impact on the $(\varepsilon, \delta)$-DP guarantee.
2. Clarify Definition 4 (Critical): Revise Definition 4 to clearly define the random projection matrix and its role in the algorithm, ensuring it is mathematically precise. In particular, with what probability does a random matrix like that satisfy the Johnson-Lindenstrauss property.
3. Justify Dynamic Noise (Critical): Explain why a decreasing noise variance is used when the privacy analysis focuses on the smallest variance.
4. Address Assumption 2: Isn't an exponent $2$ missing in the expectation ? Furthermore, I do not understand why this assumption is necessary given that we know the full noise distribution, which is Gaussian.
5. Explain Experimental Results (Recommended): Discuss why DPSGD and D2P-SGD perform similarly in experiments (e.g., Figures 2 and 3). Provide details on hyperparameter tuning, especially given that DPSGD is known to be very sensitive to a poor hyper parameters choice.
6. Highlight Convergence Proof Challenges (Recommended): If there are specific technical difficulties in the convergence proof unique to D2P2-SGD, highlight them to clarify the novelty of the analysis.

**Strengths And Weaknesses:**

Strengths:
- Timely research : Improving the privacy-utility tradeoff in private optimization is an important topic of research.
- Empirical Results: Extensive experiments across multiple datasets (FashionMNIST, SVHN, CIFAR-10) validate the method's superior accuracy compared to baselines.
- Practical Relevance: The focus on reducing noise dimensionality via random projection is well-suited for large-scale models.

Weaknesses:
- Clarity Issues: The writing could be improved for readability, particularly in definitions (e.g., Definition 4 is ambiguous and lacks clarity on the random projection matrix).
- Privacy Analysis Limitations: The privacy guarantee in Theorem 1 holds only for a restrictive upper bound on $\varepsilon$, which counterintuitively tightens with more data. The impact of random projection stochasticity on privacy is not explicitly addressed, or not clear to the reader.
- Missing Proof Details: The proof of Lemma 3 is not provided, and its source is unclear, undermining the privacy analysis.
- Experimental Clarity: The near-identical performance of DPSGD and D2P-SGD in experiments is not explained, and hyperparameter tuning details are insufficient.
- Dynamic Noise Rationale: The use of decreasing noise variance is not well-justified, especially since the privacy analysis seems to consider the worst-case (smallest) variance.

---

> ### Author Response · Authors · 2025-06-27
> **Response to Review of Paper4915 by Reviewer JbaZ**
>
> **Requested Changes**
>
> **A1:** We thank the reviewer for this comment. We have provided proof details for all necessary lemmas for proving Theorem 1 and added proper citations in the revised draft. For the stochasticity of random projection in the privacy, we have also provided two additional analyses for convex (Appendix A.5) and nonconvex (Appendix A.8) functions to shed light on how the random projection (including lower dimension and variance) affects the privacy-utility tradeoff. Specific comments have been added to the discussion after Theorem 1 as well. In general, the impact of stochasticity of random projection does not explicitly show in the privacy guarantee analysis, which has been validated in existing works (Zhou et al., 2021, Kasiviswanathan, 2021, Feng & Venkitasubramaniam, 2024) as differentially privacy is immune to the post processing.
>
> Zhou, Y., Wu, Z. S., & Banerjee, A. (2020). Bypassing the ambient dimension: Private sgd with gradient subspace identification. arXiv preprint arXiv:2007.03813.
>
> Kasiviswanathan, S. P. (2021, December). SGD with low-dimensional gradients with applications to private and distributed learning. In uncertainty in artificial intelligence (pp. 1905-1915). PMLR.
>
> Feng, C., & Venkitasubramaniam, P. (2024). Rqp-sgd: Differential private machine learning through noisy sgd and randomized quantization. arXiv preprint arXiv:2402.06606.
>
> **A2:** We appreciate this great comment from the reviewer. In the revised draft, we have turned it into a lemma (Lemma 2) based on the comment from reviewer b8zr and added clarification to make it clearer. We also followed the linear mapping that has been defined in Lemma 1 (previously Definition 3) to precisely have the form $h(o)=\frac{1}{\sqrt{p}}A^\top o$, which is called in the Algorithm 1 to project the stochastic gradient. Before and after Lemma 2, we have justified why random projection based on JL Lemma can be used in this context and give the probability that satisfies the Johnson-Lindenstrauss property (which is a well-known result in (Johnson et al., 1984)). Additionally, we have improved the writing for the improvement of readability.
>
> **A3**: We appreciate this great comment from the reviewer. The motivation for us to use a decreasing noise variance is because when keeping a constant variance, along with the optimization, this may hurt the ultimate utility. Instead, dynamic noise variance can provide more flexibility to adjust the tradeoff between privacy and utility. However, this may breach the privacy significantly if we still use the original lower bound of noise variance in the privacy analysis as stated in (Abadi et al., 2016). To keep $(\varepsilon, \delta)$-DP for the algorithm, one mitigation we adopt in this work is to correspondingly adjust the lower bound of noise variance such that even after $K$ compositions, the privacy loss will not be significant and still remain the same. This also marks one of the first a few attempts to dynamically trading off the privacy and utility, showing more flexibility in differential privacy compared to the traditional static mechanism. To further motivate our study, we have added some more discussion on this for clarification after Theorem 1.
>
> Abadi, M., Chu, A., Goodfellow, I., McMahan, H. B., Mironov, I., Talwar, K., & Zhang, L. (2016, October). Deep learning with differential privacy. In Proceedings of the 2016 ACM SIGSAC conference on computer and communications security (pp. 308-318).
>
> **A4:** We thank the reviewer for this great comment. We apologize for this typo and have corrected it in our revised draft. This assumption is generic in almost all SGD (and its variants) to characterize the negative impact of variance caused by the stochastic gradients due to mini-batch sampling. We would like to clarify that in D2P2-SGD, the full noise distribution we have added in the update is specifically for differential privacy. Besides, we also quantify the dimension distortion error due to random projection with the variance of the sampling distribution. The specific impact of these three different errors can be seen in Appendix A.8. In our revised draft, we have added more clarification on this.

---

> > ### Author Response · Authors · 2025-06-27
> > **(Continued) Response to Review of Paper4915 by Reviewer JbaZ**
> >
> > **A5:** We thank the reviewer for this great comment. The reason why DPSGD and D2P-SGD perform similarly in experiments is due to setting of the privacy. In DPSGD, the noise variance is $\sigma^2_\epsilon = \frac{C_2K\textnormal{ln}(1/\delta)B^2}{n^2\varepsilon^2}$. This corresponds to the static privacy-utility tradeoff with the fixed $\sigma^2_\epsilon$. For D2P-SGD, it has dynamic noise variance, leading to $\sigma^2_{\epsilon,k} = \frac{C_2K^2\textnormal{ln}(1/\delta)B^2}{n^2\varepsilon^2k}$. When $k\ll K$ is not large, the initialization error may dominate such that in the early phase of optimization, even if the additive noise mechanisms are different, the performance may not be diverse. When $k\to K$ is large, privacy error can dominate. However, at this stage, the noise variances between DPSGD and D2P-SGD are similar, thus resulting in similar performance. We have included more discussion into the revised draft. Regarding hyperparameter tuning details, we have added clarification in the result section and also made a table in the Appendix A.9.4.
> >
> > **A6:** We thank the reviewer for this great comment. As suggested by the reviewer here, we have listed out the difficulties in the convergence proof unique to D2P2-SGD in the Appendix (at the beginning of convex and nonconvex analysis) for the novelty of the analysis.

---

### Review · Reviewer_b8zr · 2025-06-15

**Summary Of Contributions:**

This paper presents D2P2-SGD, a new differentially private optimization framework. The method integrates three key components: a dynamic differential privacy mechanism with decaying noise variance to reduce utility loss over time, automatic gradient clipping via per-sample normalization to stabilize updates without manual tuning, and random projection to lower the dimensionality of gradients, thereby reducing the amount of added noise and computational overhead. Privacy and utility guarantees are provided for the proposed method. Empirical evaluations on benchmark datasets demonstrate that D2P2-SGD significantly improves model accuracy compared to existing DP methods.

**Audience:**

Yes

**Claims And Evidence:**

Yes

**Requested Changes:**

1. page 3, Table 1, $K$ is not defined.

2. page 4, a paragraph below Eq(1), the form of the automatic clipping mechanism seems not correct. Please check

3. Definition of $\epsilon$-DP is ot necessary since the paper focuses on $(\epsilon, \delta)$-DP

4. Definitions 3 and 4 should be written as Lemmas.

5. The explanation of random projection in the paper lacks clarity. The authors should provide a more detailed discussion, including illustrative examples or intuitive explanations to better convey how random projection contributes to the method.

6. Section 3.1 primarily focuses on the description of Algorithm 1, but the comparisons with existing methods are interwoven into this section. For better clarity and structure, the comparisons should be presented in a separate subsection or paragraph.

7. Smooth with modulus L is not defined.

8. The paragraph below Thm1, line 9 should be line 6?

9. Algorithm 1 output $x_K$, while Thm 2 focuses on $\bar{x}_K$, is there any possible to consider $x_K$?

10. There seems to be some problems with the discussion of Thm 2. There is no $\sigma_{\epsilon,k}^2$ in the bound of Thm 2. Further, Eq(10) is not given.  Besides, the tradeoff between utility and privacy should be more detailed.

11. In Appendix A.3, Thm 1, 'Assumption 2(b)' should be 'Assumption 2'.

**Strengths And Weaknesses:**

Strength: A new method D2P2-SGD is proposed by combining DPP, automatic gradient clipping via per-sample normalization, and random projection. In particular, random projection is used to lower the dimensionality of gradients, which avoids excessive random noise being added for each dimension of the gradient. Experiments show the effectiveness of the proposed method.

Weakness

1. The core idea of the paper appears to be a straightforward combination of existing techniques—specifically, the dynamic differential privacy mechanism from Bu et al. (2024) and the random projection approach from Kasiviswanathan (2021). This integration, while practical, may come across as lacking in novelty or conceptual innovation.

2. The theoretical contribution of the results is somewhat unclear. For instance, in convex optimization settings, it is well-established that DP-SGD can achieve the optimal convergence rate of $O(\frac{\sqrt{d\log(1/\delta)}}{n\epsilon})$. In contrast, Theorem 2 in this paper presents a bound that depends on additional factors such as variance, iteration count, dimensions d and p, and other parameters. It would strengthen the paper to re-express the results in this standard form to enable a direct and meaningful comparison with known bounds in the literature.

3. Another concern is the overall writing quality of the paper, which is inconsistent and contains numerous typos and grammatical issues. see below for more details.

---

> ### Author Response · Authors · 2025-06-27
> **Response to Review of Paper4915 by Reviewer b8zr**
>
> **Weakness:**
>
> **A1:** We appreciate the constructive comments from the reviewer. We respectfully counter the perception of D2P2-SGD as a straightforward combination of existing techniques from Du et al. (2021) and Kasiviswanathan (2021). While building on prior work, our framework introduces fundamental innovations: (1) The first unified integration of dynamic DP with random projection, where time-varying noise (which is different from the one used in Bu et al. (2024)) is explicitly optimized for dimension-reduced gradients, which enables synergistic privacy amplification unaddressed in isolated approaches; (2) Novel theoretical insights into the dimension-privacy-utility trilemma under convex (Theorem 2) and non-convex (Theorem 3) objectives, revealing how projection reshapes dynamic DP trade-offs; (3) A flexible mechanism-agnostic design supporting arbitrary noise/projection variants. Extensive validation confirms these co-adaptations consistently outperform conventional combinations, offer new pathways for efficient privacy-utility balancing. We believe this represents a meaningful conceptual and practical advance in scalable private learning. To underscore the significance of our proposed algorithm, we have also included this in the revised draft.
>
> **A2:** We appreciate this great comment from the reviewer. First, the optimal convergence rate $\mathcal{O}(\frac{\sqrt{d\textnormal{log}(1/\delta)}}{n\epsilon})$ essentially shows the constant factor scales with privacy and dimension. Its iteration-dependent form is $\mathcal{O}(LD(1+\frac{\sqrt{d\textnormal{log}(1/\delta)}}{n\epsilon}\cdot B)\frac{1}{\sqrt{T}})$, showing that DPSGD preserves the same $\mathcal{O}(1/\sqrt{T})$ as the non-private SGD achieves, where $L$ is the Lipschitz constant, $D$ is the diameter of the constraint set, which typically is the squared norm difference between the initialization and the optimal solution, and $B=\mathcal{O}(1)$ is the batch size. Therefore, to enable a direct comparison, we also convert our bound to such a form that is with respect to the privacy and dimension. It should be noted that our proposed method integrates random projection and dynamic differential privacy together, which results in a more complex error bound. Recalling the injected noise variance in Theorem 1, i.e., $\sigma^2_\epsilon$ and substituting into the error bound in Theorem 2, with the step size $\alpha=\frac{p^{3/2}}{d^2\sqrt{K}}$, we can rewrite the upper bound as $\mathcal{O}(\frac{1}{\sqrt{K}}\cdot(\frac{Dd^2}{p^2\sigma^2_A}+\frac{\sqrt{p\textnormal{ln}(1/\delta)}}{d^2n\epsilon}+p^{5/2}\sigma^2_A))$, when $C_2=\frac{n\epsilon}{p^{5/2}K\textnormal{ln}K\sqrt{\textnormal{ln}(1/\delta)}}$. Compared to the originally optimal convergence rate of vanilla DPSGD, the second term suggests that the privacy error is reduced due to the random projection and the dynamic privacy mechanism. However, this comes at the cost of approximation error caused by the random projection as well. Additionally, the optimization error is also scaled by the dimensions. Therefore, this rewritten bound reflects on the optimization-privacy-dimension tradeoff. We have included this comparison in the Appendix of the revised draft (A.5 for convex and A.8 for nonconvex). We still have the regular iteration-dependent rate in the main text as done in (Koloskova et al., 2023).
>
> Koloskova, A., Hendrikx, H., & Stich, S. U. (2023, July). Revisiting gradient clipping: Stochastic bias and tight convergence guarantees. In International Conference on Machine Learning (pp. 17343-17363). PMLR.
>
> **A3:** We thank the reviewer for this great comment and have revised the draft based on the comments in the requested changes.
>
> **Requested Changes:**
>
> **A1:** We have defined $K$ in the revised draft.
>
> **A2:** For the automatic clipping mechanism, we followed Eq.(3.5) in (Bu et al., 2024), where they further showed in Eq.(4.1) that any constant choice of $G>0$ is equivalent to using $G=1$. Therefore, in Algorithm 1, we set $G=1$ and multiply it by the gradient, which is also similar to line 4 in Algorithm 1 in (Bu et al., 2024). To maintain the consistency, we has multiplied it by $\mathbf{v}$ and made the corresponding change in the draft.
>
> **A3:** As suggested here, we have removed the definition of $\epsilon$-DP and only focus on $(\epsilon,\delta)$-DP.
>
> **A4:** As suggested by the reviewer, we have turned Definitions 3 and 4 into lemmas in the revised draft.
>
> **A5:** We really appreciate the comment from the reviewer. In the revised draft, we have added additional intuitive explanations in Section 3.1 on how random projection contributes to the method.
>
> **A6:** In the revised draft, we have created a separate paragraph for the comparisons with existing methods.
>
> **A7:** In Assumption 1, we have defined the smoothness modulus for completeness in the revised draft.
>
> **A8:** We appreciate the comment from the reviewer to correct this typo. In our revised draft, we have made the change.

---

> > ### Author Response · Authors · 2025-06-27
> > **(Continued) Response to Review of Paper4915 by Reviewer b8zr**
> >
> > **A9:** We thank the reviewer for this comment. It is possible to consider it instead of the average. We can use $\frac{1}{K}\sum_{k=1}^K\mathbb{E}[f(\mathbf{x}_k)-f^*]$ in our convergence error analysis. However, the one in Theorem 2 has genetically been used in convergence analysis of convex optimization problems, such as in (Garrigos & Gower, 2023). This is due to the summation over $1,2,…,K$ in the analysis. Therefore, we follow it in our draft.
> >
> > Garrigos, G., & Gower, R. M. Handbook of convergence theorems for (stochastic) gradient methods (2023). arXiv preprint arXiv:2301.11235.
> >
> > **A10:** We thank the reviewer for this comment. The reason why $\sigma^2_{\epsilon,k}$ is missing is because based on its definition $\sigma_{\epsilon,k}=\frac{\sigma_\epsilon}{\sqrt{k}}$, we have summarized it over 1, 2,…, K such that it becomes $1 + 1/2+ 1/3 + … + 1/K \leq \textnormal{ln}K + 1$.We can observe that this enables the second term of the bound in Theorem 2. Substituting the upper bound of $\sigma^2_\epsilon$ also results in the explicit impact of privacy and dimension on the bound, which has been detailed in the response to Comment 2 in the Strengths and Weaknesses. We have also corrected the equation numbering issue due to the crossing references. According to the suggestion from the reviewer, we have elaborated the tradeoff between utility and privacy in the revised draft.
> >
> > **A11:** We appreciate this comment from the reviewer and have corrected this in the revised draft.

---

### Review · Reviewer_uWeG · 2025-06-15

**Summary Of Contributions:**

This paper studies differentially private optimizer with focus on the trade-off between privacy, utility, and complexity. The authors propose D2P2-SGD algorithm, which combines the idea of random projection and dynamic noise variance. The method naturally encompasses the existing two methods. They provide theoretical results of the proposed method, including privacy bound and utility bound. Through empirical results, the proposed methods is shown to outperform other baselines when strong privacy guarantee does not needed.

**Audience:**

Yes

**Claims And Evidence:**

Yes

**Requested Changes:**

I would request the authors to address my concerns listed in the Strengths and Weaknesses section above.

**Strengths And Weaknesses:**

The paper is easy to follow and well written. The proposed method seems reasonably designed and the authors support their method with both theoretical and empirical results. Although the method shows better performance in the empirical setting of the paper, I have some questionable things:
1. How does Definition 3 support the use of random projection in DPSGD? In particular, why the preservation of distance is important?
2. How does the random mapping in Definition 4 provide the mapping $h$ in Definition 4? What is the probability guarantee happening it?
3. Performance comparison (e.g. Figure 2, 3) under the same $\epsilon$ setting should be made. Is the proposed method performing better than baselines with the same $\epsilon$?
4. (minor) In section 3.1, line 5 or line 7 seem to indicate different sentences in the algorithm 1. For example Line 9 in the algorithm 1 is return $x_K$.

---

> ### Author Response · Authors · 2025-06-27
> **Response to Review of Paper4915 by Reviewer uWeG**
>
> 1. How does Definition 3 support the use of random projection in DPSGD? In particular, why the preservation of distance is important?
>
> We appreciate the useful comment from the reviewer. We would like to point out that in our work, we do not directly apply Definition 3 in our algorithm design and analysis. Instead, Definition 3 serves as a motivation for us to apply random projection to model parameters. Since random projection based on JL Lemma was originally proposed for data and the preservation of distance is a defining feature of the lemma. Without it, random projection would be a lossy and unreliable heuristic. The lemma provides a mathematical guarantee that geometry survives dimensionality reduction, enabling scalable and accurate computation. When we apply random projection to model parameters to achieve dimensionality reduction, it aims to find an effective projection (i.e., linear map) that can also follow JL lemma. Thus, Definition 3 actually offers a justification why we can apply random projection to model parameters like it can be applied to data and underscores the significance of linear map defined in Definition 4. We have added clarification in the revised draft.
>
> 2. How does the random mapping in Definition 4 provide the mapping $h$ in Definition 4? What is the probability guarantee happening it?
>
> We thank the reviewer for this useful comment. In Definition 4, we didn’t really explicitly define what $h$ is. We apologize for this confusion and have made change to keep the consistency in the revised draft, i.e., $h(o)=r=\frac{1}{\sqrt{p}}A^\top o$ in Definition 4, where $A$ is a random matrix with elements being sampled from a Gaussian distribution, while $o$ is a vector. The probability guarantee of using the linear mapping $h$ with the support from JL Lemma is at least $1-\frac{1}{m}$, where $m$ is the size of any set of points with a certain dimension. Mirroring this in a parameter space, we can know that the probability guarantee is similar, while the probability has now become $1-\frac{1}{n}$, where $n$ is the size of private dataset $\mathcal{D}$. We have also changed the definitions (Definition 3 and Definition 4 to Lemma 1 and Lemma 2) to lemmas based on the comment from another reviewer.
>
> 3. Performance comparison (e.g. Figure 2, 3) under the same $\epsilon$ setting should be made. Is the proposed method performing better than baselines with the same $\epsilon$?
>
> We thank the reviewer for this great comment. We have included additional results in the revised draft to show the performance comparison under the same $\epsilon$ (Table 2). The empirical results show that the proposed method performs better than baselines with the same $\epsilon$.
>
> 4. (minor) In section 3.1, line 5 or line 7 seem to indicate different sentences in the algorithm 1. For example Line 9 in the algorithm 1 is return $\mathbf{x}_K$.
>
> We thank the reviewer for this comment. We have corrected all line numbering issues in Algorithm 1 in the revised draft.

---

### Author Response · Authors · 2025-06-27
**Response to all reviewers**

We sincerely thank all reviewers for their constructive and insightful comments, which have significantly strengthened our work. Each specific concern raised has been carefully addressed in the revised manuscript (changes highlighted in blue). We appreciate the time and expertise invested in evaluating our paper and welcome any further feedback.

---

### Decision · Action_Editor_VUEz · 2025-08-23

**Recommendation:** Accept as is

**Additional Comments:**

In this paper, this paper proposed D2P2-SGD algorithm which is supported by theory on the trade-off between utility and privacy and experimental results. The core idea of the paper appears to be a straightforward combination of existing techniques—specifically, the dynamic differential privacy mechanism from Bu et al. (2024) and the random projection approach from Kasiviswanathan (2021). While the novelty may not be significant, the paper does contain new ideas and results which are of interest to TMLR audience. Two reviewers recommended acceptance. one reviewer recommended rejection but they still acknowledge that the proposed method seems reasonably designed and the authors support their method with both theoretical and promising empirical results.

Considering the acceptance recommendation from two reviewers who are experts in the area, I recommend acceptance.

**Audience:**

Yes

**Audience Explanation:**

Differentially privacy is one appealing topic in machine learning and it certainly attract interests in TMLR.

**Claims And Evidence:**

Yes

**Claims Explanation:**

the proposed methods are valdated by experiments.